# Non-autonomous insulin signaling delays mitotic progression in *C. elegans* germline stem and progenitor cells

Eric Cheng, Ran Lu, Abigail R. Gerhold *

Department of Biology, McGill University, Montréal, Canada

* abigail.gerhold@mcgill.ca

## Abstract

Stem and progenitor cell mitosis is essential for tissue development and homeostasis. How these cells ensure proper chromosome segregation, and thereby maintain mitotic fidelity, in the complex physiological environment of a living animal is poorly understood. Here we use *in situ* live-cell imaging of *C. elegans* germline stem and progenitor cells (GSPCs) to ask how the signaling environment influences stem and progenitor cell mitosis *in vivo*. Through a candidate screen we identify a new role for the insulin/IGF receptor (IGFR), *daf-2*, during GSPC mitosis. Mitosis is delayed in *daf-2*/IGFR mutants, and these delays require canonical, DAF-2/IGFR to DAF-16/FoxO insulin signaling, here acting cell non-autonomously from the soma. Interestingly, mitotic delays in *daf-2*/IGFR mutants depend on the spindle assembly checkpoint but are not accompanied by a loss of mitotic fidelity. Correspondingly, we show that caloric restriction, which delays GSPC mitosis and compromises mitotic fidelity, does not act via the canonical insulin signaling pathway, and instead requires AMP-activated kinase (AMPK). Together this work demonstrates that GSPC mitosis is influenced by at least two genetically separable signaling pathways and highlights the importance of signaling networks for proper stem and progenitor cell mitosis *in vivo*.

**Data Availability Statement:** All numeric data reported in this manuscript have been deposited at Dryad (https://doi.org/10.5061/dryad.sn02v6xfp), except for the data used in S2 Fig, which is available in S1 Data. Live-cell imaging data,

## Author summary

Stem and progenitor cells drive tissue development and sustain adult tissue turnover by producing new daughter cells via cell division, the success of which depends on proper chromosome segregation during mitosis. Stem and progenitor cells perform mitosis in the complex environment of a living animal, yet relatively little is known about how events during mitosis are influenced by this *in vivo* context. In particular, whether signaling pathways that coordinate other aspects of stem and progenitor cell behavior with animal physiology also play a role during mitosis is poorly understood. Here we took advantage of the germline stem and progenitor cells of the model nematode *C. elegans* to address this question. Through live-cell imaging of germline stem and progenitor cell mitosis, we uncover a new role for the insulin signaling pathway. We find that reducing insulin signaling delays germline stem and progenitor cell mitosis, but, surprisingly, these delays are not accompanied by a loss of mitotic fidelity. In addition, we find that reducing insulin

collected during the candidate screen presented in Fig 1, have been deposited at the Federated Research Data Repository (FRDR; https://doi.org/10.20383/103.01149).

**Funding:** This work was funded by grants to ARG from the Fonds de recherche du Québec Nature et technologies (FRQNT grant 283252) and the Natural Sciences and Engineering Research Council of Canada (NSERC grant RGPIN/05199-2020). The funders had no role in study design, data collection and analysis, decision to publish, or preparation of the manuscript.

**Competing interests:** The authors have declared that no competing interests exist.

signaling in somatic tissues is sufficient to delay germline stem and progenitor mitosis, indicating that the pathway acts non-autonomously. Finally, while insulin signaling is known to link cell division with nutritional status in many species, we found that it did not mediate the effects of caloric restriction on germline stem and progenitor cell mitosis. Instead, caloric restriction acts via the conserved energy-sensing regulator AMPK. These results uncover new regulators of germline stem and progenitor cell mitosis and emphasize the importance of signaling pathways for proper stem and progenitor cell mitosis *in vivo*.

## Introduction

The ability of stem and progenitor cells to support tissue development and homeostasis relies on the proper execution of events in mitosis within the changeable environment of a living animal. Signaling pathways allow stem and progenitor cells to respond to their environment, often by coupling cell cycle progression with organismal physiology and developmental cues [1–3]. Whether these pathways also impact events in mitosis, particularly those required for stem and progenitor cells to ensure proper chromosome segregation and therefore genomic integrity, is less clear.

During mitosis, the biorientation of replicated chromosome pairs on the mitotic spindle, via stable kinetochore-microtubule attachments, ensures that, upon chromosome segregation in anaphase, each daughter cell inherits exactly one copy of the replicated genome. The spindle assembly checkpoint safeguards this process by delaying anaphase in the presence of unattached kinetochores [4]. Unattached kinetochores recruit checkpoint proteins to catalyze formation of the mitotic checkpoint complex (MCC), which prevents the anaphase-promoting complex/cyclosome (APC/C) from targeting mitotic substrates, like cyclin B and securin, for degradation. Defects in spindle assembly and/or checkpoint surveillance can lead to chromosome segregation errors, compromising genomic stability and cellular fitness [5].

Signaling pathways can influence spindle assembly and/or checkpoint activity directly, by regulating the activity of mitotic proteins, or indirectly, through changes in gene expression. These changes can be maladaptive with the net effect of making mitosis more error prone. For example, in transformed human cells, hyperactive Ras/MAPK signaling leads to excessive phosphorylation of kinetochore proteins and impaired kinetochore-microtubule attachments [6], while changes in insulin signaling can adversely affect the expression of mitotic genes in several cell types [7–10]. Alternatively, signaling events may protect mitotic fidelity. For example, AMP-activated protein kinase (AMPK) signaling allows cells to avoid spindle defects and mitotic delays by increasing energy production during mitosis [11]. However, much of the work linking signaling pathways to events in mitosis has been carried out in *ex vivo* model systems. Consequently, how the signaling environment affects mitosis *in vivo*, particularly in stem and progenitor cells, is not well understood.

The germline stem and progenitor cells (GSPCs) of the nematode *Caenorhabditis elegans* are an excellent model system for studying stem and progenitor cells mitosis *in vivo*. *C. elegans* adult hermaphrodites have two pools of GSPCs, one located at the distal tip of each of the two U-shaped gonad arms (Fig 1A; reviewed in [12]). GSPCs are kept in an undifferentiated, mitotic state by proximity to a somatic niche, the distal tip cell. Collectively, the distal region of the germ line is referred to as the progenitor or proliferative zone and comprises two populations of mitotically cycling germ cells—the distal-most cells with the highest level of niche signaling are considered stem cells. These cells divide symmetrically with respect to cell fate and

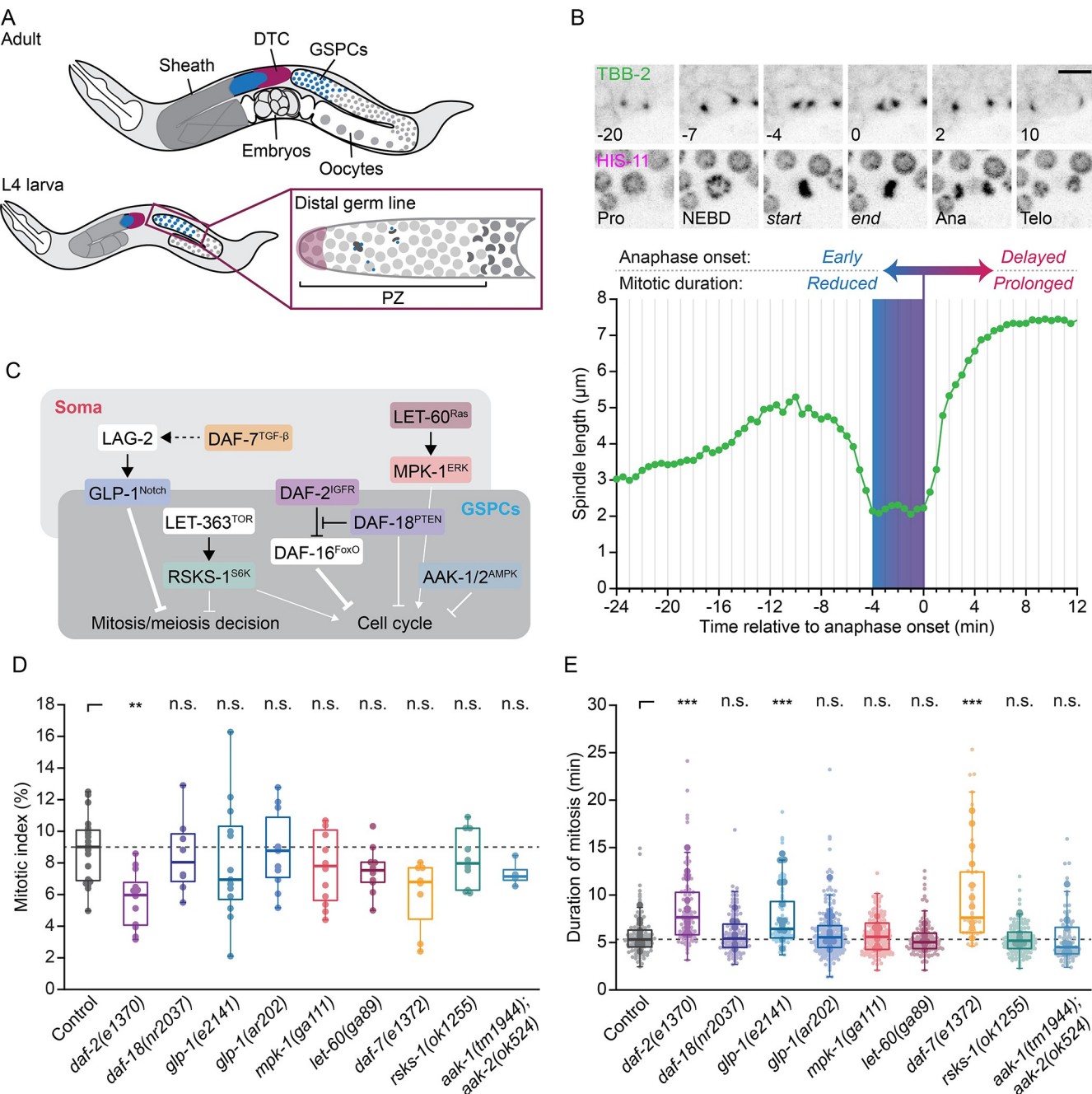

**Fig 1. Mutations in *daf-2*/IGFR, *daf-7*/TGF-β and *glp-1*/Notch delay GSPC mitosis.** (A) Cartoon representation showing the organization of the *C. elegans* somatic gonad and germ line in adults (top) and L4 larvae (bottom), with a close up of the distal proliferative zone (PZ), which houses GSPCs. (B) Single time point, maximum intensity projections of GFP::TBB-2 (top) and HIS-11::mCH (bottom) during GSPC mitosis, with a plot of spindle length over time for the cell shown below. Numbers indicate time in minutes, relative to the metaphase-anaphase transition, for different mitotic phases or key transitions. The metaphase-anaphase transition (anaphase onset) is indicated on the plot by a purple line. Early anaphase onset is associated with reduced mitotic durations (blue) while a delayed anaphase onset is associated with prolonged mitotic duration (red). Pro = Prophase, NEBD = nuclear envelope breakdown, start = first time point after NEBD when spindle length stabilizes, end = anaphase onset, Ana = Anaphase, Telo = Telophase. Scale bar = 5 μm. (C) Schematic illustrating the role of conserved signaling pathways in regulating GSPC fate and/or proliferation and whether they act in the soma or germ line. Major points of regulation are indicated by line weight. (D-E) The PZ mitotic index (D) and GSPC duration of mitosis (E) in L4 larvae bearing mutant alleles in the signaling pathways shown in (C). The dashed grey line indicates the median value for control. The mitotic index is lower in *daf-2(e1370)* animals and mitosis is delayed in *daf-2(e1370)*, *glp-1(e2141)* and *daf-7(e1372)* animals. In (D), dots represent one PZ and one PZ was assessed per animal. In (E), small dots represent individual GSPCs, larger dots represent the mean value per gonad arm/PZ. In both (D) and (E), boxplots show the median, interquartile range and most extreme values not considered statistical outliers. n.s. = $p > 0.05$, ** = $p < 0.01$, *** = $p < 0.001$. Summary statistics and statistical tests used for all figure panels are given in S2 Data. See also S1 and S2 Figs.

are maintained according to a population-based model [13]. The more proximal cells are progenitor cells completing their final mitotic cell cycle before entering meiosis [14]. Meiotic germ cells can become nurse cells, eventually undergoing apoptosis, or give rise to mature gametes, which are found at the proximal end of the gonad.

GSPCs proliferate extensively during larval development, driving a more than thousandfold increase in the number of germline cells, and continue to divide in reproductively active adults to support gamete production (Fig 1A). While several signaling pathways are known to impact cell cycle progression in GSPCs, aligning the rate of GSPC proliferation with tissue demand and animal physiology [15–21], relatively little is known about how GSPCs maintain the fidelity of mitosis and thus the quality of these divisions. Here we took advantage of the fact that GSPC mitosis can be observed directly by *in situ* live-cell imaging [22–24], to assess how mitosis is influenced by signaling pathway activity in the whole-animal context. We report a candidate genetic screen for signaling pathway mutants that impact mitotic duration in GSPCs and uncover a role for the insulin/IGF receptor (IGFR) *daf-2* in promoting proper GSPC mitotic timing. We show that DAF-2/IGFR acts via the canonical insulin/IGF-1 signaling (IIS) pathway, converging on the transcription factor DAF-16/FoxO; however, the predominant site of pathway activity resides outside of the germ line in the soma, and is thus cell non-autonomous. Surprisingly, we find that while reduced IIS leads to checkpoint-dependent mitotic delays, it does not compromise mitotic fidelity. This is different from caloric restriction, which delays GSPC mitosis [22], but which we show here also increases the frequency of chromosome segregation errors when surveillance by the spindle assembly checkpoint is removed. Correspondingly, we find different genetic requirements for mitotic delays downstream of caloric restriction and reduced DAF-2/IGFR activity. Together, our results demonstrate that mitosis in GSPCs is sensitive to the signaling environment and suggest a degree of specificity in terms of how different signaling pathways interact with the core mitotic machinery.

## Results

### A candidate screen for signaling pathways that affect the duration of mitosis uncovers *daf-2*/IGFR

In GSPCs, the duration of mitosis is regulated by the spindle assembly checkpoint, with conditions that disrupt spindle assembly or weaken the checkpoint prolonging or reducing mitotic duration, respectively (Fig 1B; [22]). We therefore reasoned that we could use the duration of mitosis to identify signaling pathways whose activity is required for proper spindle assembly and/or checkpoint activity in GSPCs. To measure the duration of GSPC mitosis, we perform live-cell imaging and use germline-expressed, fluorescent protein(FP)-tagged β-tubulin (FP::TBB-2) to track pairs of mitotic spindle poles, which allows us to monitor mitotic progression by looking for the stereotypical changes in spindle length that accompany mitotic transitions (Fig 1B; [22–24]). We define the duration of mitosis as the amount of time between the end of nuclear envelope breakdown (NEBD), when spindle length stops decreasing, until the metaphase-anaphase transition, after which spindle length increases rapidly. Spindle pole tracking also allows us to measure spindle length and dynamics throughout mitosis (S2A Fig; hereafter "spindle features" [24]), which can be indicative of defects in spindle assembly and/or chromosome alignment and segregation (e.g. [25–28]). We focused our analysis on the L4 larval stage of development (Fig 1A), as GSPCs proliferate extensively during this time and may be more sensitive to changes in environmental and/or developmental signaling [3,29,30].

We targeted conserved signaling pathways that regulate GSPC fate (i.e. whether to remain mitotic or enter meiosis) and/or cell cycle progression, both during development and in the adult animal (Fig 1C; reviewed in [3,12]). The Notch pathway regulates the mitosis-meiosis

decision in GSPCs [31], but does not affect cell cycle progression per se [14,29]. Transforming Growth Factor-β (TGF-β) signaling acts both upstream of and in parallel to Notch signaling to regulate GSPC fate during larval development [32,33], and has also been implicated in the maintenance of GSPCs during reproductive aging [34]. The TOR pathway is thought to affect both GSPC fate and cell cycle progression during larval development [35,36] and plays a role in regulating GSPC quiescence during L1 larval and dauer diapause [37–39]. Regulation of GSPC quiescence in larvae and adults also requires AMP-activated kinase (AMPK) [16,38,40] and the lipid and protein phosphatase PTEN, here acting independently of the transcription factor FoxO [15–17,37,41,42]. Ras/MAPK pathway activity affects GSPC proliferation in larvae and adults [40,43,44], although its primary role in the germline is to promote meiotic progression and oocyte maturation [43,45,46]. Finally, the IIS pathway, via inhibition of FoxO, promotes GSPC proliferation and is particularly important for developmental expansion of the GSPC pool [17,29,47].

We crossed mutant alleles of key genes in each pathway into our reference strain, which carries FP::TBB-2 (here GFP::TBB-2) to mark spindles and mCherry-tagged histone H2B (mCH::HIS-11) to mark chromatin (Fig 1B). To validate these strains, we first scored the GSPC mitotic index, by counting the total number of proliferative zone (PZ) nuclei and the total number of cells in mitosis (Figs 1D, S1A and S1B). We found that reducing insulin pathway activity via a hypomorphic allele of the insulin/IGF receptor (IGFR) *daf-2* (*daf-2(e1370)*; [48]; hereafter *daf-2(rf)*) lead to a lower mitotic index, while all other conditions were not significantly different from control, despite wide ranging alterations in the number of PZ nuclei. This result aligns well with previous reports suggesting that the insulin pathway is the chief regulator of GSPC cell cycle progression during larval development [29]. The mitotic index also tended to be lower in animals carrying a hypomorphic allele of *daf-7*/TGF-β (*daf-7 (e1320)*; [49]), although this result was not statistically different from control (*p* = 0.09), suggesting that TGF-β signaling may also contribute to GSPC proliferation. In our hands, a putative null allele of ribosomal S6 kinase (S6K) *rsks-1* (*rsks-1(ok1255)*; [50]), which acts downstream of TOR in many species, and which has been associated with fewer PZ cells [35], was not significantly different from controls (S1A Fig). We suspect that this discrepancy may be due to differences in genetic background, but additional experimentation is required to test this directly.

To determine whether these mutations impact the duration of mitosis in GSPCs, we performed live-cell imaging and spindle pole tracking. We found that GSPCs in *daf-2(rf)* and *daf-7(e1320)* animals, and in animals bearing a conditional loss-of-function allele of *glp-1*/Notch (*glp-1(e2141)*; [51]) were delayed in mitosis, while mitotic duration in other mutant backgrounds was no different from control (Fig 1E). To determine whether mitotic delays in different mutant backgrounds were linked to similar changes in spindle assembly, we compared spindle features across all genotypes (S2 Fig). We found that strains with mitotic delays were unlike in both the magnitude and directionality of change for most spindle features (S2B Fig). For example, fluctuations in spindle length during spindle assembly were reduced in *daf-2(rf)*, increased in *glp-1(e2141)* and unchanged in *daf-7(e1320)*, relative to controls (S2B Fig). Correspondingly, no single spindle feature predicted the duration of mitosis across all cells (S2C Fig). To consider all spindle features at once, we performed a principal component analysis (PCA). While our data did not segregate into well-defined clusters, delayed cells in *daf-2(rf)* and *daf-7(e1320)* tended to separate in PC space from those in *glp-1(e2141)* animals (S2D Fig), further suggesting that mitotic delays in these mutant backgrounds are not caused by a common set of spindle defects. Altogether, our data indicate that GSPC mitosis is sensitive to changes in the IIS, TGF-β and Notch pathways and suggest a degree of specificity in how each pathway impacts spindle formation and/or dynamics.

### *daf-2*/IGFR regulates mitotic duration via the canonical IIS pathway

To begin to decipher how different signaling pathway mutants affect GSPC mitosis, we focused our attention on *daf-2(rf)* and the IIS pathway. DAF-2/IGFR activates a canonical phosphoino-sitide 3-kinase (PI3K)/Akt signaling cascade, which is antagonized by DAF-18/PTEN, to inhibit the Forkhead family transcription factor DAF-16/FoxO (Fig 2A). This core pathway is

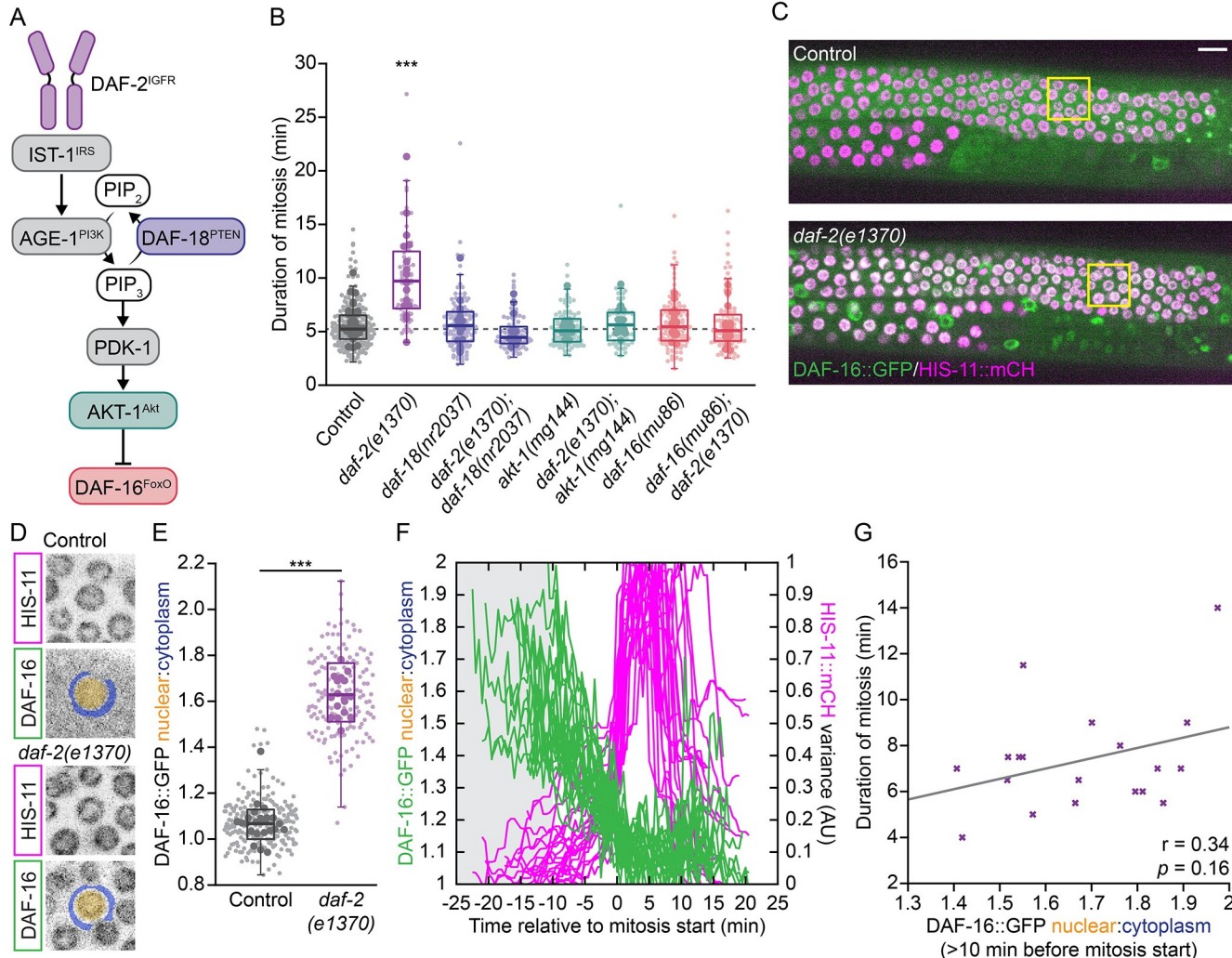

**Fig 2. *daf-2(e1370)* delays GSPC mitosis via the canonical IIS pathway converging on DAF-16/FoxO.** (A) Schematic showing the core IIS pathway with pathway components color-coded as in (B). (B) The duration of GSPC mitosis in animals bearing the mutant alleles or allele combinations indicated. *daf-2 (e1370)* mitotic delays are suppressed by loss-of-function mutations in *daf-18* and *daf-16* and a gain-of-function mutation in *akt-1*. The dashed grey line indicates the median value for control. *daf-2(e1370)* is delayed relative to all other genotypes. All other genotypes are not different from control. (C) Maximum intensity projections of the top half of a gonad arm from control (top) and *daf-2(e1370)* (bottom) L4 larvae expressing DAF-16::GFP, from the endogenous *daf-16* locus, and HIS-11::mCH in germ cells. Yellow boxes indicate the cells shown in (D). (D) Maximum intensity projections of single control (top) and *daf-2(e1370)* (bottom) GSPC nuclei. Blue and yellow overlays show the regions used to measure DAF-16::GFP cytoplasmic and nuclear fluorescence intensity, respectively. (E) The DAF-16::GFP nuclear-to-cytoplasmic ratio (N:C) is elevated in *daf-2(e1370)* GSPCs relative to control. (F) Changes in the DAF-16::GFP N:C (green) and HIS-11::mCH variance (magenta) over time relative to mitosis start for *daf-2(e1370)* GSPCs. Grey shaded region shows the period of time prior to the start of mitosis when DAF-16::GFP was measured for mitotic cells. Mitotic duration was inferred from changes in the variance of HIS-11::mCH. (G) DAF-16::GFP N:C prior to mitosis does not predict the duration of mitosis on a per cell (x) basis. Grey line shows the linear least-squares best fit for the data, with Pearson's coefficient (r) and *p* value shown below. For (B) and (E), small dots represent individual GSPCs, larger dots represent the mean value per gonad arm/PZ. Boxplots show the median, interquartile range and most extreme values not considered statistical outliers. *** = *p* < 0.001. Summary statistics and statistical tests used for all figure panels are given in S2 Data. See also S3 Fig.

conserved and plays a major role in integrating cell proliferation with organismal physiology in many species (reviewed in [52]).

In *C. elegans*, reducing IIS leads to fewer GSPCs, which has been attributed to a DAF-16/ FoxO-dependent delay in the G2 phase of the cell cycle during larval expansion of the GSPC pool [29]. Correspondingly, we found that the low GSPC mitotic index in *daf-2(rf)* mutant larvae was suppressed by a catalytically dead allele of *daf-18*/PTEN (*daf-18(nr2037)* [53]; hereafter *daf-18(lf)*) and a dominant-activating allele of *akt-1* (*akt-1(mg144)* [54]; hereafter *akt-1(gf)*), which increase pathway activity downstream of DAF-2/IGFR, and by a null allele of *daf-16*/ FoxO (*daf-16(mu86)* [55]; hereafter *daf-16(0)*), which simulates the transcriptional response induced by high pathway activity (S3A Fig, [29]).

To test whether the canonical IIS pathway also mediates the effect of *daf-2(rf)* on the duration of mitosis in GSPCs, we used live-cell imaging and spindle pole tracking to measure mitotic duration in *daf-18(lf)*, *akt-1(gf)* and *daf-16(0)* single mutants and in double mutants carrying each of these alleles with *daf-2(rf)* (Fig 2B). We found that the duration of mitosis was unchanged in *daf-18(lf)*, *akt-1(gf)* and *daf-16(0)* single mutants, as compared to wild-type control, suggesting that elevating IIS is not sufficient to accelerate basal mitotic timing; however, all three mutations fully suppressed *daf-2(rf)* mitotic delays, indicating that reduced DAF-2/ IGFR activity prolongs GSPC mitosis via the canonical IIS pathway converging on the regulation of DAF-16/FoxO.

DAF-2/IGFR inhibits DAF-16/FoxO transcriptional activity by regulating its subcellular localization: when DAF-2/IGFR is active, DAF-16/FoxO is sequestered in the cytoplasm, while reduced DAF-2/IGFR activity allows for DAF-16/FoxO to accumulate in the nucleus [56–58]. Accordingly, if DAF-2/IGFR regulates GSPC mitosis via DAF-16/FoxO-dependent changes in gene expression we would expect increased DAF-16/FoxO nuclear localization in GSPCs when DAF-2/IGFR activity is reduced. To test this, we measured the nuclear-to-cytoplasmic ratio of DAF-16::GFP in GSPCs, using a strain in which the endogenous *daf-16*/FoxO locus bears a carboxyl-terminal GFP tag to mark all isoforms (DAF-16::GFP; [59]) (Fig 2C). We found that DAF-16::GFP was enriched in GSPC nuclei in both *daf-2(rf)* animals (Fig 2C–2E) and in animals in which *daf-2* was depleted by RNAi (S3B Fig), suggesting that a systemic reduction in insulin signaling leads to elevated DAF-16/FoxO activity in GSPCs.

In *daf-2(rf)* animals, the duration of GSPC mitosis varies from ~5 minutes, which is similar to the mean duration for controls, up to ~25 minutes (see for example Figs 1E **and** 2B). To determine whether differences in DAF-16::GFP nuclear enrichment might account for this variation, we plotted the DAF-16::GFP nuclear-to-cytoplasmic ratio prior to mitosis versus mitotic duration on a per cell basis (Figs 2F–2G and S3C). However, we found only a weak, non-statistically significant, correlation between these two values. Furthermore, nuclear DAF-16::GFP was elevated (nuclear-to-cytoplasmic ratio > 1.4) even in *daf-2(rf)* GSPCs with relatively normal durations of mitosis (4–6 min). Thus, while reducing the activity of DAF-2/ IGFR leads to increased nuclear localization of DAF-16/FoxO in GSPCs, nuclear enrichment of DAF-16/FoxO in GSPCs does not appear to be sufficient to generate mitotic delays.

## IIS delays GSPC mitosis cell non-autonomously

The lack of a predictive relationship between GSPC DAF-16/FoxO nuclear enrichment and GSPC mitotic duration prompted us to ask whether DAF-2/IGFR to DAF-16/FoxO signal transduction was acting in GSPCs (i.e. cell autonomously) to promote proper mitotic timing or whether cell non-autonomous signaling might be involved. To test this idea, we used the auxin-inducible degron (AID) system which allows for the conditional depletion of AID-tagged proteins using tissue-specific expression of the F-box substrate recognition component

TIR1 [60]. We used a strain in which the endogenous *daf-2*/IGFR locus is tagged at the carboxyl-terminus with AID and mNeonGreen (DAF-2::AID::mNG; [61]). We first checked the expression of DAF-2::AID::mNG in the larval germ line and found that it was present at low levels in the PZ, appearing largely as cytoplasmic puncta, with weak membrane localization in GSPCs (S4A Fig).

We generated strains carrying DAF-2::AID::mNG and a germline-specific TIR1 (*sun1p*::*TIR1*; [60]), with mCherry-tagged β-tubulin (mCH::TBB-2) for spindle pole tracking. DAF-2::AID::mNG levels were lower in the PZ of these animals even without supplemental auxin, suggesting some leaky degradation (S4B and S4C Fig), while auxin treatment led to robust DAF-2/IGFR depletion in the germline, reducing DAF-2::AID::mNG levels in the PZ below our threshold of detection (Fig 3A and 3B). Surprisingly, germline DAF-2/IGFR depletion did not delay GSPC mitosis; rather, we observed a marginal, but statistically significant, decrease in the duration of mitosis (Fig 3C). These results imply that mitotic delays in *daf-2(rf)* animals may be due to a cell non-autonomous requirement for DAF-2/IGFR activity. Correspondingly, depletion of *daf-2/IGFR* using germline-specific RNAi [62] did not affect mitotic duration in GSPCs, whereas RNAi depletion in the whole animal induced significant mitotic delays (S4D Fig).

As reducing germline DAF-2/IGFR was not sufficient to delay GSPC mitosis, we next tested whether DAF-2/IGFR might be acting in the soma. We performed analogous experiments using a strain bearing a pan-somatic TIR1 (*eft-3p*::*TIR1* with the *unc-54* 3' UTR; [60]). *eft-3p*::*TIR1* DAF-2::AID::mNG animals showed reduced DAF-2::AID::mNG in somatic tissues and phenotypes consistent with a partial loss of DAF-2/IGFR somatic activity without auxin treatment, again suggesting leaky degradation (Figs 3A and S4E). However, germline expression of DAF-2::AID::mNG, was no different from control (S4B and S4C Fig) and higher than what we observed in germline-specific TIR1 animals without auxin treatment (Figs 3A, 3B, S4B and S4C), suggesting that germline DAF-2/IGFR was intact. We therefore took advantage of these animals to test the effect of partial somatic DAF-2/IGFR depletion on GSPC mitosis and found that mitosis was significantly delayed (Fig 3C). Together these results imply that the primary focus of DAF-2/IGFR activity responsible for GSPC mitotic delays resides in the soma.

To further pinpoint the somatic tissues involved, we used the AID system to deplete DAF-2::AID::mNG from neurons (*rgef-1p*::*TIR1*), muscle (*myo-3p*::*TIR1*), intestine (*ges-1p*::*TIR1*), hypodermis (*dpy-7p*::*TIR1*) and somatic gonadal sheath cells (*lim-7p*::*TIR1*) [61]. Depletion from the intestine or gonadal sheath cells was sufficient to delay mitosis in GSPCs, while depletion from other tissues was not (Fig 3C). Further, depletion of DAF-2::AID::mNG from the somatic sheath cells and intestine did not reduce DAF-2::AID::mNG levels in the germline (S4B and S4C Fig). Taken together, our data strongly suggest that DAF-2/IGFR acts cell non-autonomously to affect mitotic timing in GSPCs, and that, of the tissues tested, the intestine and gonadal sheath cells are the principal foci of DAF-2/IGFR activity.

To determine whether DAF-16/FoxO also acts cell non-autonomously, we first assessed whether germline (*sun1p*::*TIR1*, with and without auxin) and/or partial somatic (*eft3p*::*TIR1*, no auxin) depletion of DAF-2/IGFR affected the localization of DAF-16::GFP in GSPCs. We found that DAF-16::GFP was enriched in GSPC nuclei in all three conditions relative to control, but was highest when DAF-2/IGFR was depleted from the germ line by auxin treatment (S5A Fig), consistent with a germline-autonomous role for DAF-16/FoxO. However, since GSPC mitosis is not delayed when DAF-2/IGFR is depleted from the germline, yet under these conditions nuclear DAF-16::GFP is at its highest, these results further suggest that nuclear localization of DAF-16/FoxO in GSPCs does not play a major role in determining the duration of GSPC mitosis.

To formally test where DAF-16/FoxO activity is required for GSPC mitotic delays, we used the AID system to deplete DAF-16/FoxO (DAF-16::GFP::AID; [61]) in the germ line (*sun-1p*::

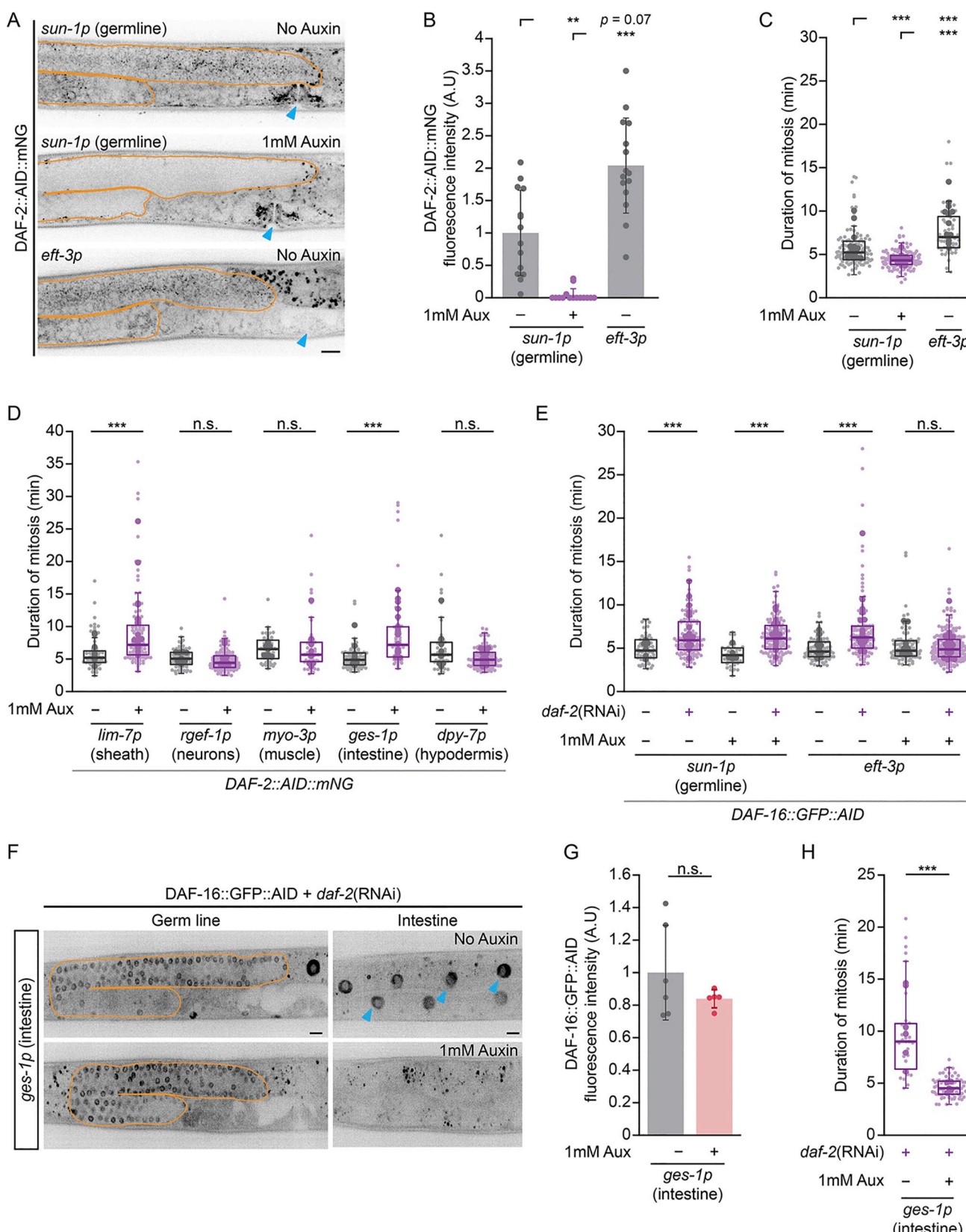

**Fig 3. The IIS pathway acts cell non-autonomously from the soma to influence GSPC mitosis.** (A) Maximum intensity projections showing DAF-2::AID::mNG in the germline (outlined in orange) and developing vulva (blue arrow head) in animals carrying *sun-1p::TIR1* with (middle) and without (top) auxin treatment, and *eft-3p::TIR1* (bottom) without auxin treatment. The vulva is shown as an example of somatic depletion in *eft-3::TIR1*, but not in *sun-1p::TIR1*, animals. Scale bar = 10 μm. (B-C) The mean DAF-2::AID::mNG fluorescence intensity per distal germ line, normalized to *sun-1p::TIR1* no auxin (B), and the duration of GSPC mitosis (C), in *sun-1p::TIR1* animals with and without auxin, and *eft-3p::TIR1* animals without auxin. Auxin treatment of *sun-1p::TIR1* animals depletes DAF-2::AID::mNG in the germ line, but does not delay GSPC mitosis. DAF-2::AID::mNG germline expression in *eft-3p::TIR1* no auxin animals is similar to *sun-1p::TIR1* no auxin, yet GSPC mitosis is delayed. (D) The duration of GSPC mitosis in animals after DAF-2::AID::mNG was depleted in different somatic tissues. Depletion in the somatic gonadal sheath cells (*lim-7p*) or intestine (*ges-1p*) was sufficient to delay GSPC mitosis. (E) The duration of GSPC mitosis in animals after *daf-2* was depleted by RNAi (purple) compared to control (grey), with and without auxin treatment to deplete DAF-16::GFP::AID in the germ line (*sun-1p::TIR1*) or in all tissues (*eft-3p::TIR1*). DAF-16/FoxO depletion in the germ line does not suppress GSPC mitotic delays induced by *daf-2*(RNAi), while *eft-3p::TIR1*-driven depletion of DAF-16, which affects both soma and germ line (S5B Fig), does. (F) Single z-slice sections through the middle of the germ line (orange outline; left), or a basal region through intestinal nuclei (blue arrow heads; right) in the same animals, showing DAF-16::GFP::AID expression in *daf-2*(RNAi) treated animals. Auxin treatment of *ges-1p::TIR1* animals does not change germline expression of DAF-16::GFP::AID, but leads to robust intestinal depletion. Scale bar = 10 μm. (G) Quantification of germline DAF-16::GFP::AID fluorescence for the animals assayed in (H). Data is normalized to *ges-1p::TIR1* no auxin. (H) The duration of GSPC mitosis in animals after *daf-2* was depleted by RNAi, and with or without auxin treatment to deplete DAF-16::GFP::AID in the intestine (*ges-1p::TIR1*). DAF-16/FoxO depletion in the intestine suppresses GSPC mitotic delays induced by *daf-2*(RNAi). For (B) and (G), dots represent the mean value per each gonad arm and one gonad arm was assessed per animal. Bar plots show the mean with error bars showing the standard deviation. For (C-E) and (H), small dots represent individual GSPCs, larger dots represent the mean value per gonad arm/PZ. Boxplots show the median, interquartile range and most extreme values not considered statistical outliers. For all plots, n.s. = $p > 0.05$, ** = $p < 0.01$, *** = $p < 0.001$. Summary statistics and statistical tests used for all figure panels are given in S2 Data. See also S4 and S5 Figs.

*TIR1*) or in the soma (*eft-3p::TIR1*) and asked whether either was sufficient to suppress GSPC mitotic delays induced by whole-animal RNAi knockdown of *daf-2*. In these experiments, *eft-3p::TIR1* induced significant depletion of DAF-16::GFP::AID in both the soma and the germ line following auxin treatment, while *sun-1p::TIR1*-driven depletion was restricted to the germ line as expected (S5B Fig). However, auxin treatment only suppressed *daf-2*(RNAi)-induced GSPC mitotic delays in the *eft-3p::TIR1* background (Fig 3E), consistent with the notion that DAF-16/FoxO, like DAF-2/IGFR, acts primarily in the soma. To corroborate this result, we performed the same experiment, but this time depleting DAF-16::GFP::AID in the intestine. Here, germline DAF-16::GFP::AID expression was unaffected (Fig 3F), yet *daf-2*(RNAi)-induced GSPC mitotic delays were fully suppressed (Fig 3G). Thus, we conclude that IIS, from DAF-2/IGFR to DAF-16/FoxO, acts in the soma to delay GSPC mitosis, with the intestine serving as a major site of pathway activity.

## Reduced IIS delays mitosis via the spindle assembly checkpoint but does not compromise mitotic fidelity

To better understand how IIS affects mitosis in GSPCs we first tested for the involvement of the spindle assembly checkpoint. We used a null allele of the core checkpoint gene *mdf-2*/Mad2 (*mdf-2(lt4)* [63]; hereafter *mdf-2(0)*) to ablate checkpoint activity and assessed GSPC mitotic duration in *mdf-2(0)* single and *daf-2(rf)*; *mdf-2(0)* double mutant animals. The duration of mitosis in *mdf-2(0)* GSPCs was significantly reduced compared to control, consistent with checkpoint regulation of basal mitotic timing in these cells [22], In *daf-2(rf)*; *mdf-2(0)* double mutants, mitotic delays were fully suppressed and the duration of GSPC mitosis was no different than in *mdf-2(0)* single mutants (Fig 4A), indicating that mitotic delays are entirely dependent on an intact spindle assembly checkpoint.

Checkpoint-dependent mitotic delays are typically due to defects in spindle assembly, and correspondingly, accompanied by chromosome segregation errors when checkpoint surveillance is removed. To determine whether reduced DAF-2/IGFR impairs spindle assembly, we asked whether *daf-2(rf)*; *mdf-2(0)* GSPCs were more prone to mitotic errors. We used live-cell imaging and scored GSPCs for mitotic errors using mCH::HIS-11 to monitor chromosome alignment and segregation (Fig 4B and 4C). The frequency of mitotic errors was low in both

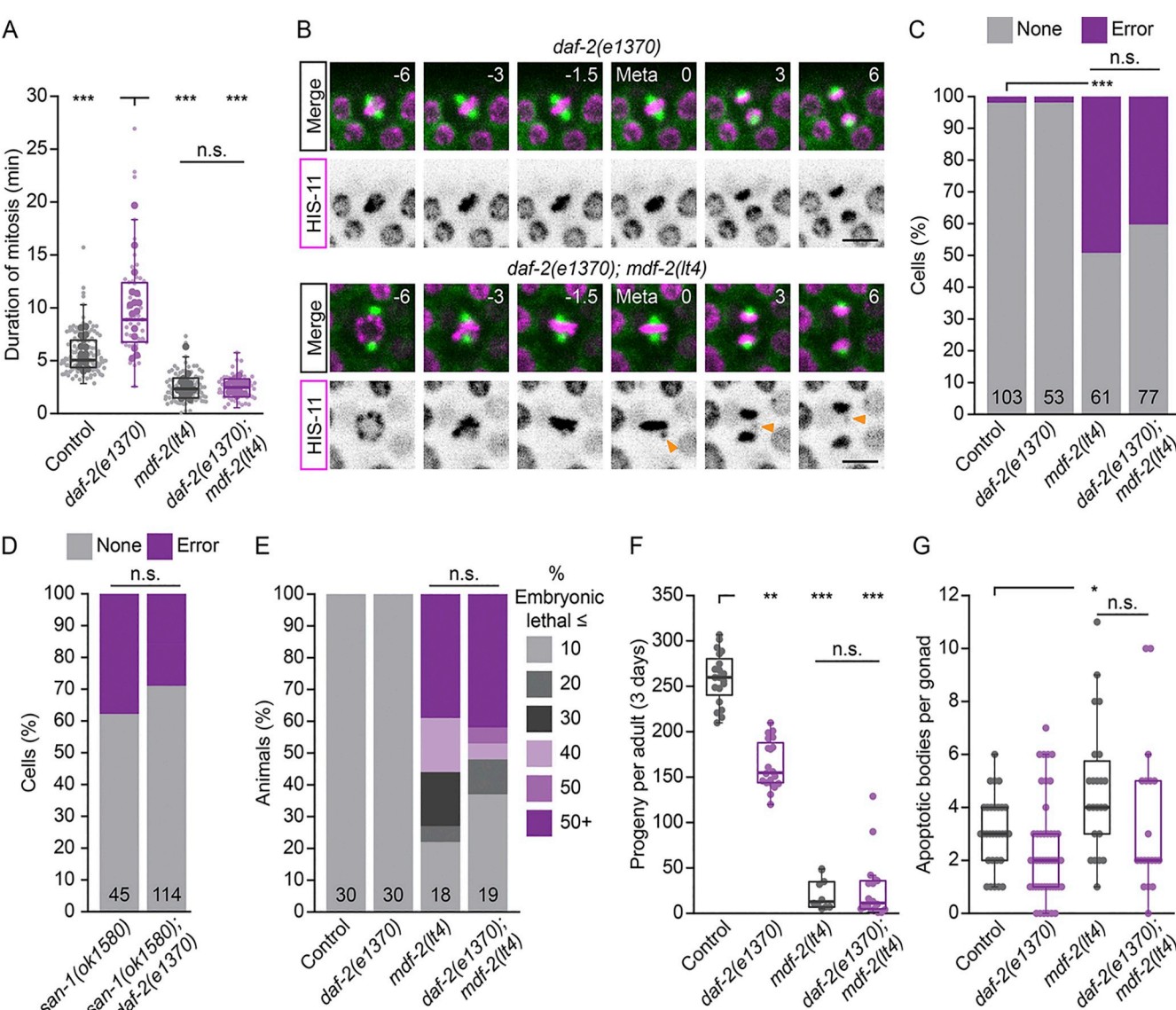

**Fig 4. Reduced IIS leads to checkpoint-dependent mitotic delays without compromising mitotic fidelity.** (A) The duration of GSPC mitosis in *daf-2 (e1370)* and *mdf-2(lt4)* single mutants and *daf-2(e1370); mdf-2(lt4)* double mutants compared to control. Loss of the checkpoint, via *mdf-2(lt4)*, suppresses *daf-2(e1370)* mitotic delays entirely. Small dots represent individual GSPCs, larger dots represent the mean value per gonad arm/PZ. (B) Single time point, maximum intensity projections of GFP::TBB-2 (green) and HIS-11::mCH (magenta) during GSPC mitosis, with the HIS-11::mCH channel shown in inverted grey scale below. Numbers indicate time in minutes, relative to anaphase onset. Orange arrowhead indicates a chromosome that fails to align at the metaphase plate and segregates improperly in anaphase. Scale bar = 5 μm. (C) The percent of cells showing mitotic errors, as shown in (B), for *daf-2(e1370)* and *mdf-2 (lt4)* single mutants and *daf-2(e1370); mdf-2(lt4)* double mutants as compared to control. The total number of cells scored for each genotype is shown at the base of each stacked bar plot. *daf-2(e1370)* does not increase the frequency of mitotic errors in *mdf-2(lt4)* mutants. (D) The percent of cells showing mitotic errors for *san-1(ok1580)* single mutants and *daf-2(e1370); san-1(ok1580)* double mutants. The total number of cells scored for each genotype is shown at the base of each stacked bar plot. *daf-2(e1370)* also does not increase the frequency of mitotic errors in *san-1(ok1580)* mutants. (E) Percent embryonic lethality (dead embryos/total embryos laid) as assessed for individual animals. Animals were binned according to the percent embryonic lethality shown on the right. The number of animals scored per genotype is shown at the base of each stacked bar plot. (F) The total number of progeny produced over a 3 day period for individual animals. (G) The number of apoptotic bodies per gonad arm. *daf-2(e1370)* does not increase the severity *mdf-2(lt4)* embryonic lethality, low brood size and elevated germline apoptosis. For (F) and (G), dots represent the value for one animal (F) or one gonad arm (G), with one gonad arm assessed er animal. For (A), (F) and (G), boxplots show the median, interquartile range and most extreme values not considered statistical outliers. For all plots, n.s. = $p > 0.05$, * = $p < 0.05$, ** = $p < 0.01$, *** = $p < 0.001$. Summary statistics and statistical tests used for all figure panels are given in S2 Data.

control and *daf-2(rf)* GSPCs, suggesting that if spindle assembly is compromised in *daf-2(rf)* cells, checkpoint-dependent mitotic delays are sufficient to ensure these cells achieve biorientation prior to anaphase onset. In *mdf-2(0)* GSPCs we observed significantly more mitotic errors, as would be expected for checkpoint-null cells, but surprisingly, the frequency of mitotic errors was not significantly different between these cells and *daf-2(rf); mdf-2(0)* double mutants. We obtained similar results using putative null allele of a second checkpoint gene *san-1*/Mad3 (*san-1(ok1580)* [50]; hereafter *san-1(lf)*) (Fig 4D). Correspondingly, several known checkpoint-null phenotypes that are associated with germline function were not enhanced in *daf-2(rf); mdf-2(0)* animals, including embryonic lethality and reduced brood size [64] (Fig 4E and 4F), nor did we observe increased germ cell apoptosis, which clears nuclei with DNA damage and/or chromosome abnormalities [65–67] (Fig 4G). Altogether, we conclude that reducing DAF-2/IGFR activity leads to checkpoint-dependent mitotic delays that are not associated with defects in spindle assembly.

### Reduced IIS and caloric restriction delay mitosis via different pathways

The effects of reduced IIS on GSPC mitosis are different from what we have previously reported for caloric restriction. Caloric restriction leads to GSPC mitotic delays that depend on the spindle assembly checkpoint; however, caloric restriction also enhances several phenotypes associated with increased chromosome segregation errors when the checkpoint is impaired [22]. We verified that caloric restriction compromises mitotic fidelity by scoring the frequency of mitotic errors in animals fed a 1 in 10 dilution of their normal bacteria food source (sDR; [68]) with and without checkpoint regulation, as done for *daf-2(rf)*. We found that sDR increased the frequency of mitotic errors in *san-1(lf)* GSPCs, as compared to *san-1(lf)* GSPCs in animals fed *ad libitum* (Fig 5A). Thus, unlike the mitotic delays caused by reduced IIS, mitotic delays upon caloric restriction reflect defects in spindle assembly.

We next tested the genetic requirements for GSPC mitotic delays upon caloric restriction, focusing on members of the IIS pathway. In addition to sDR, we also used a mutation in *eat-2* (*eat-2(ad465)*; hereafter *eat-2(rf)*)) that slows pharyngeal pumping and reduces food intake [69] to induce caloric restriction. We found that *daf-16(0)*, which fully suppressed mitotic delays in *daf-2(rf)* animals (Fig 2B), did not suppress mitotic delays in either *eat-2(rf)* or sDR-treated animals (Fig 5B and 5C). These results indicate that canonical DAF-2/IGFR to DAF-16/FoxO signaling does not mediate the effect of caloric restriction on GSPC mitosis and, by extension, downstream effectors other than DAF-16/FoxO must be involved.

*daf-18*/PTEN plays both *daf-16*/FoxO-dependent and independent roles in regulating GSPC proliferation during development and adulthood [15–17,29,37,42]. We found that *daf-18(lf)* restored normal mitotic timing in both *eat-2(rf)* and sDR-treated animals (Fig 5B and 5C). In several situations where *daf-18*/PTEN acts independently of *daf-16*/FoxO to regulate GSPC behavior, it acts upstream of, or in parallel to, AMPK [38,40]. We tested for AMPK involvement using animals bearing null mutations in both catalytic α subunits (*aak-1(tm1944)* and *aak-2(ok524)* [50,70,71]; hereafter *aak-1/2(0)*). Like *daf-18(lf)*, *aak-1/2(0)* also rescued GSPC mitotic delays upon sDR (Fig 5D). Based on these results, we conclude that while caloric restriction and reduced IIS both delay GSPC mitosis, they do so via distinct mechanisms.

### Discussion

Here we have investigated the signaling requirements for proper GSPC mitosis as a model for how mitosis in stem and progenitor cells is regulated *in vivo*, in the whole-animal context. Through our candidate screen we found that reducing the activity of *daf-2*/IGFR, *daf-7*/TGF-β or *glp-1*/Notch prolongs the duration of mitosis in GSPCs. As only a subset of the mutant

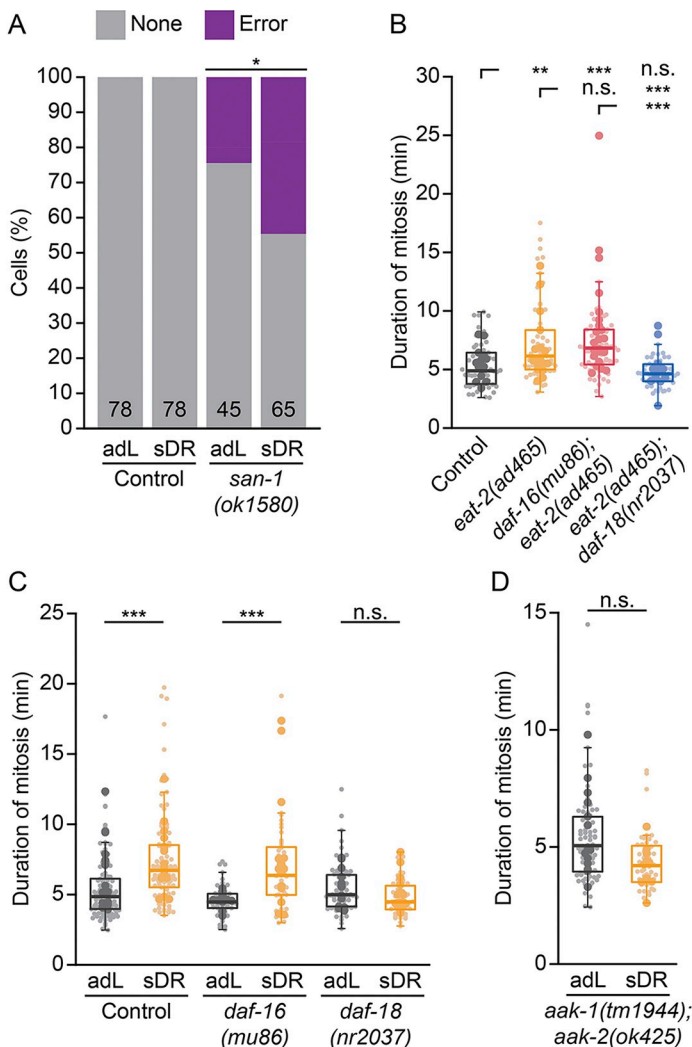

**Fig 5. Reduced IIS and caloric restriction rely on different downstream signaling effectors to delay GSPC mitosis.**
(A) The percent of cells showing mitotic errors for animals fed *ad libitum* (adL) or subjected to caloric restriction (sDR) in control versus *san-1(ok1580)* single mutants. The total number of cells scored for each condition is shown at the base of each stacked bar plot. Unlike *daf-2(e1370)*, sDR increases the frequency of mitotic errors in *san-1(ok1580)* mutants. (B-D) The duration of GSPC mitosis in *eat-2(ad465)* single mutants compared to control, *daf-16(mu86)*; *eat-2(ad465)* and *eat-2(ad465); daf-18(nr2037)* double mutants (B), in control versus *daf-16(mu86)* and *daf-18(nr2037)* single mutant adL and sDR animals (C) and in AMPK null animals (*aak-1(tm1944); aak-2(ok425)*) fed *ad libitum* or subjected to sDR (D). GSPC mitotic delays upon caloric restriction depend on *daf-18* and *aak-1/2*, but not *daf-16*. Small dots represent individual GSPCs, larger dots represent the mean value per gonad arm/PZ. Boxplots show the median, interquartile range and most extreme values not considered statistical outliers. For all plots, n.s. = $p > 0.05$, ** = $p < 0.01$, *** = $p < 0.001$. Summary statistics and statistical tests used for all figure panels are given in S2 Data.

backgrounds in our screen delayed GSPC mitosis, we think it unlikely that these effects are a generic response to perturbed signaling compromising animal health. Our data also suggest that delayed cells from different mutant backgrounds exhibit different combinations of spindle features, which may indicate specificity in terms of the mitotic processes that are affected. This idea is further supported by our results showing that mitotic delays in *daf-2*/IGFR mutants and in animals subjected to caloric restriction can be suppressed genetically, but rely on different downstream signaling effectors and are associated with different phenotypes in terms of mitotic fidelity. Finally, while the absolute magnitude of these delays is small (on the order of

minutes), they are significant given the normal duration of GSPC mitosis ($< 6$ min, on average), and similar in magnitude to those seen following targeted perturbation of kinetochore-microtubule attachments [22]. Together, these results argue that the changes we have observed in the duration of GSPC mitosis are likely the product of specific interactions between signaling pathway effectors and mitotic events. Whether these changes are adaptive (e.g. increasing checkpoint activity to buffer cells from environmental fluctuations) or indicative of mitotic vulnerabilities (e.g. changes in gene expression or protein activity that compromise spindle assembly) requires additional investigation.

To start to address this question, we investigated the nature and source of mitotic delays in *daf-2*/IGFR mutants. We found that mitotic delays in *daf-2*/IGFR mutants require downstream signaling components, including the transcription factor *daf-16*/FoxO, suggesting that canonical DAF-2/IGFR-to-DAF-16/FoxO IIS is involved. However, we also showed that these delays depend on DAF-2/IGFR and DAF-16/FoxO activity in the soma, indicating that IIS affects GSPC mitosis cell non-autonomously. We further showed that while GSPC mitotic delays in *daf-2*/IGFR mutants depend on the spindle assembly checkpoint, in the absence of checkpoint regulation, these cells are no more likely to experience chromosome segregation errors than cells in an unperturbed insulin signalling environment, ruling out defects in kinetochore-microtubule attachment as the source of mitotic delays. Finally, we demonstrated that GSPC mitotic delays induced by caloric restriction do not depend on DAF-16/FoxO suggesting that canonical IIS is not responsible for coordinating GSPC mitosis with nutritional status. Here we discuss the implications of these results focusing on the possible causes of GSPC mitotic delays when IIS is reduced and the signaling mechanism by which this information is transduced.

The absence of any overt defects in spindle assembly in *daf-2*/IGFR mutants raises the question as to why GSPCs are delayed in mitosis? As these delays are entirely dependent on the spindle assembly checkpoint, it is unlikely that they are due to events following checkpoint satisfaction (e.g. a slower rate of APC/C-directed degradation of cyclin B and/or Securin should slow mitotic progression with or without checkpoint activity [72]). Instead, we think it more likely that reduced IIS affects the checkpoint itself, by enhancing checkpoint activity and/or delaying checkpoint silencing. For example, reduced insulin signalling could increase the strength of checkpoint signaling at individual kinetochores relative to their attachment state [73,74] or increase the microtubule occupancy threshold for checkpoint silencing [75–79]. Alternatively, the rate of MCC turnover in the cytoplasm could be slower [80–86], favoring MCC accumulation and greater APC/C inhibition. Experiments to assess checkpoint strength and kinetochore enrichment of checkpoint proteins in GSPCs when IIS is reduced will help to differentiate between these possibilities and clarify the mechanism by which mitosis is delayed. A role for IIS in modulating checkpoint strength would be particularly interesting as it might serve to protect cells from environmental or physiological conditions that could compromise mitotic fidelity and would therefore be adaptive. Identifying other conditions that like caloric restriction delay GSPC mitosis, but which also depend on DAF-16/FoxO will be necessary to address this hypothesis.

In considering the nature of the changes to GSPC mitosis downstream of IIS, we also have to account for the fact that the insulin pathway is acting cell non-autonomously from the soma. Most strikingly, depleting DAF-2/IGFR from the intestine was sufficient to delay GSPC mitosis, like in *daf-2(rf)* mutants, while depleting DAF-16/FoxO in the intestine fully suppressed delays caused by systemic RNAi knockdown of *daf-2*/IGFR. Depleting DAF-2/IGFR from the somatic gonadal sheath cells was also sufficient to delay GSPC mitosis and we did not test whether depleting DAF-16/FoxO in these cells was sufficient to suppress *daf-2*(RNAi)-induced mitotic delays. In addition, while expression of *ges-1p::TIR1*

and *lim-7p*::*TIR1* should be specific to the intestine and somatic sheath cells, respectively [61], we cannot guarantee that no other somatic cells are affected. Therefore, while our results are consistent with a predominant role for intestinal IIS, we do not exclude the possibility that IIS in several somatic tissues may contribute to changes in the duration of GSPC mitosis.

Our results do, however, rule out a role for germline-autonomous IIS. We found that germline depletion of DAF-2/IGFR did not delay GSPC mitosis and germline depletion of DAF-16/FoxO did not rescue mitotic delays when *daf-2*/IGFR was knocked down by RNAi. Conversely, depletion of either DAF-2/IGFR or DAF-16/FoxO in the intestine strongly affected the duration of mitosis in GSPCs despite robust germline expression of both proteins. Finally, while DAF-16/FoxO is expressed in GSPCs and was enriched in GSPC nuclei when *daf-2*/IGFR activity was reduced, this enrichment does not correlate with mitotic duration either on a per cell basis or when considered across treatments. For example, germline depletion of DAF-2/IGFR lead to an increase in nuclear DAF-16::GFP, but did not delay GSPC mitosis.

This result was somewhat surprising given that the rate of GSPC proliferation during larval development is thought to be regulated by germline-autonomous IIS [12,29]. However, other aspects of GSPC behavior are controlled by somatic IIS [34,87,88], and somatic signaling via several other pathways influences GSPC proliferation and/or differentiation throughout development and adulthood [21,30,32,37,42,44,89,90], providing ample precedent for non-autonomous regulation of GSPCs. In addition, of the three signaling pathways uncovered by our screen, only IIS plays a major role in regulating GSPC proliferation [14,29,32,33,91], suggesting that mitotic and cell cycle timing can be uncoupled and may be regulated by distinct mechanisms. Accordingly, we find that mitosis is also delayed in GSPCs in *daf-2(rf)* adults (S6A Fig), even though the impact of *daf-2(rf)* on GSPC proliferation is limited to larval development (S6B Fig; [29,47]). Altogether, we think it most likely that different mechanisms underlie the influence of IIS on GSPC cell cycle versus mitotic timing with cell non-autonomous pathway activity governing mitosis.

Cell non-autonomous IIS occurs when DAF-16/FoxO activity in one tissue influences IIS and DAF-16/FoxO activity in a second tissue, so-called FoxO-to-FoxO signaling [92,93], or when DAF-16/FoxO-independent responses are induced in the signal receiving tissue [94,95]. As germline DAF-16/FoxO was not necessary to delay GSPC mitosis we favor the latter mechanism. The lack of a cell autonomous role for DAF-16/FoxO also rules out the possibility that DAF-16/FoxO-dependent changes in GSPC gene expression determine GSPC mitotic timing. Thus, unlike Forkhead family transcription factors in yeast and human cells, which affect the timing of mitotic events and mitotic fidelity by regulating the expression of mitotic genes [7,96–98], DAF-16/FoxO appears to influence mitotic timing in *C. elegans* GSPCs solely via its involvement in inter-organ communication.

Based on our tissue-specific DAF-2/IGFR depletion experiments, the intestine and somatic sheath cells are the likely sources for a DAF-16/FoxO-dependent secondary signal that acts on GSPCs. In *C. elegans*, the intestine comprises the entire endoderm and serves as a major hub for homeostatic regulation [99]. The somatic sheath cells, which enwrap the germ line and are connected to GSPCs by gap junctions for most of larval development, support larval GSPC proliferation [30,100–103]. Thus, both tissues are well positioned to regulate GSPC mitosis cell non-autonomously. Determining the identity of the secondary signal which acts on GSPCs and how it interfaces with mitosis will provide a framework for understanding how inter-organ communication contributes to mitotic regulation.

## Materials and methods

### *C. elegans* strains and culture

*C. elegans* strains were maintained at either 15°C or 20°C on nematode growth medium (NGM), seeded with *Escherichia coli* (*E. coli*) strain OP50 (NGM/OP50), following standard protocols [104]. See S1 Table for the full list of strains used in this study. For live-cell imaging, animals were synchronized at the L1 larval or embryonic stage and raised at 20°C, at a density of 30–40 larvae per plate. Larvae were raised on standard NGM plates seeded with the *E. coli* strain HT115 (NGM/HT115), which produces more uniform developmental timing and a more consistently high number of GSPC divisions [23,105]. Late L4 larvae were selected 2–4 days later, depending on synchronization method and genetic background. For all experiments using thermosensitive alleles (*daf-2(e1370)*, *glp-1(e2141)*, *glp-1(ar202)*, *mpk-1(ga111)*, *let-60(ga89)* and *daf-7(e1372)*), animals were raised at the semi-permissive temperature of 20°C from the L1 stage. For a subset of the data included in the candidate screen and for the *daf-2*(RNAi) experiments presented in S3B and S4D Figs, synchronized L1 larvae were obtained by sodium hypochlorite treatment [106]. Briefly, gravid adults were washed from the surface of a standard NGM/OP50 plate using 1 ml M9 buffer (22 mM $KH_2PO_4$, 42 mM $Na_2HPO_4$, 86 mM NaCl, 1 mM $MgSO_4$), and dissolved in 1 ml of a solution of 1% NaClO and 500 mM NaOH in water, for no more than 7 min, with regular agitation. Released embryos were rinsed and incubated for 24hrs at 15°C in M9 with rotation. For data presented in Figs 2E–2G and 4A, strains were maintained on NGM/OP50 plates by weekly transfer and small "clumps" of approximately L1 staged larvae were transferred by worm pick 1–3 days after the bacterial lawn had been fully consumed. For the majority of the data included in the candidate screen and all other experiments, synchronized animals were obtained by egg collection. Briefly, 3–5 adults were transferred to an NGM/HT115 plate and let to lay eggs for 2–3 hrs, after which, adults were removed.

### Auxin treatment for AID depletion

35 mm NGM plates containing 1 mM 1-naphthaleneacetic acid, potassium salt (K-NAA), a synthetic auxin analog that is photostable and water soluble [107] (Phytotechnology Laboratories N610) were seeded with 150 μl of HT115 and incubated at room temperature overnight. Control plates with no K-NAA were prepared in parallel. For experiments using TIR1 expressed from the *sun-1* (germ line), *rgef-1* (neurons), and *lim-7* (gonadal somatic sheath cells) promoters and DAF-2::AID::mNG, animals were raised from hatching on 1 mM K-NAA plates. In our hands, strains carrying TIR1 expressed from the *ges-1* (intestine), *myo-3* (muscle), or *dpy-7* (hypodermis) promoters and DAF-2::AID::mNG entered dauer when raised on 1 mM K-NAA plates. To avoid dauer entry, animals were raised on control plates for 24 hrs, after which they were transferred to 1 mM K-NAA or fresh control plates. For experiments using DAF-16::GFP::AID, animals were raised from hatching on 1 mM K-NAA or control plates. We note that weak depletion of AID-tagged proteins in animals raised on control plates (i.e. without supplemental auxin) was evident in several of our experiments (*eft-3p*::TIR1 with DAF-2::AID::mNG and *sun-1p*::TIR1 with DAF-2::AID:: mNG). Similar leaky degradation has been documented elsewhere [108,109]. We also found that *eft-3p*::TIR1, which was designed to express TIR1 in the soma and not in the germ line [60], lead to robust germline depletion of DAF-16::GFP::AID, suggesting that *eft-3p*::TIR1 is at least weakly expressed in the germline and that this is sufficient to drive degradation of some AID-tagged proteins.

## RNAi depletion of *daf-2*

RNAi was performed by feeding, as described previously [110] with the following modifications. A plasmid for the expression of double-stranded RNAi targeting *daf-2* was constructed by cloning exon 9 of *daf-2* into the L4440 vector and transforming into the HT115 *E. coli* strain. Exon 9 of *daf-2* was amplified with Phusion High-Fidelity DNA polymerase (New England Biolabs, M0530S) using primers F_Primer: 5'-TAAGCACCATGGCACCTGTAC CTTCTCCTTCAAC-3' and R_Primer: 5'-TGCTTAGGGCCCCCTCTTCGATCGTCATGT TCTC-3'. The amplicon and L4440 vector were digested using ApaI (New England Biolabs, R0114S) and NcoI (New England Biolabs, R0193S) restriction enzymes. Ligation was performed using T4 DNA ligase (New England Biolabs, M0202S) and the resulting plasmid was transformed into the HT115 *E. coli* strain following standard procedures. HT115 carrying the *daf-2* plasmid or the empty vector L4440 were cultured for 12–16 hrs in LB + 100 μg/ml ampicillin (LB/Amp) at 37˚C. This culture was diluted 1:100 in LB/Amp and grown for another 6 hrs under the same conditions, after which 100 μl was added per 35 mm NGM/IPTG/Carb plate (NGM with 1 mM isopropyl β-d-thiogalactoside (IPTG) and 50 μg/ml carbenicillin). Seeded plates were kept in the dark, at room temperature for 3 days. For experiments combining *daf-2(RNAi)* with AID depletion of DAF-16, plates were prepared identically, except that 1 mM K-NAA was added to the NGM/IPTG/Carb plates at the plate pouring stage.

## Dietary Restriction (sDR)

35 mm plates containing standard NGM, with 50 μg/ml carbenicillin to prevent additional bacterial growth, were seeded with 150 μl of HT115. A fresh overnight HT115 culture was used either undiluted (control) or after a 1:10 dilution (sDR) in S basal (1M NaCl, 0.44M $KH_2PO_4$, 0.06M $K_2HPO_4$, 0.05mg/mL Cholesterol). Plates were dried at room temperature for 24 hrs. Synchronous cultures were obtained by egg collection, after rinsing adult animals in M9 buffer to reduce bacterial transfer.

## Animal mounting and live-cell timelapse and single time point imaging

Animals were mounted for live-cell imaging and/or single time point imaging as described in [23]. Briefly, animals were anesthetized in 0.04% tetramisole in M9 buffer and transferred to a 3% agarose pad, molded using a silicon wafer etched to imprint square-shaped grooves 35 μm deep, with widths ranging from 38–55 μm. Animals were positioned in grooves by gently blowing air through a mouth pipet or using a worm eyelash pick. An 18 mm 1.5 coverslip was lowered gently, the corners were sealed using VaLaP (1:1:1 Vaseline, lanolin, and paraffin) and the area under the coverslip was backfilled with M9 + 0.04% tetramisole. For whole-animal imaging of DAF-2::AID::mNG, animals were mounted following the same procedure except that 3% agarose pads without grooves were used. Images were acquired at room temperature (∼20˚C) with either a Quorum WaveFX-X1 spinning disk confocal (Leica DMI6000B inverted microscope, with an ASI MS-2000 piezo stage, controlled by MetaMorph software, using a Leica 63x/1.40–0.60 HCX PL APO oil immersion objective, 491 nm (50 mW) and 568 nm (50 mW) diode lasers, with ET 525/50 and FF 593/40 emission filters, using dual camera mode and two Photometrics PRIME BSI sCMOS cameras), or a Nikon CSU-X1 spinning disk confocal (Nikon TI2-E inverted microscope with a Yokogawa CSU-X1 confocal scanner, controlled by NIS-Elements software, using a Nikon Apo 40x/1.25 NA water immersion objective, 488 nm (100mW) and 561 nm (100mW) solid-state lasers, with a dual band pass Chroma 59004m filter, and a Hamamatsu ORCA-Fusion BT sCMOS camera). Specific imaging parameters are listed by experiment in S2 Table.

## Image processing and analysis

All image processing and analysis was carried out in Fiji [111], using a combination of in-house, semi-automated scripts and manual approaches, as specified below. Data were compiled and analyzed in MATLAB (Mathworks), except where noted.

## Measuring mitotic duration and spindle features using spindle pole tracking

Spindle pole tracking and subsequent data analysis were performed as described in [22–24] with minor modifications. In Fiji, images were converted from NIS-Elements or MetaMorph file formats into a standard hyperstack TIF, with image metadata exported to a CSV file. Animal movement was corrected prior to spindle pole tracking by XY registration, as described in [23,24]. Spindle pole tracking was performed using the TrackMate plug-in [112,113]. Tracks were manually curated in TrackMate to create a relatively gap-free, single track per spindle pole, and all spots in each track were renamed to indicate appropriate pairing (i.e. spindle poles belonging to the same cell). The TrackMate 'Spots' information for each germ line was exported as a TXT file and the XYZT coordinates and labels for all tracked spindle poles were imported into MATLAB for scoring and analysis. Pole-to-pole distance (spindle length) was plotted relative to time and NEBD, the "start" of mitosis, and anaphase onset were scored by clicking on spindle length versus time plots, as previously described [22–24]. If plots were ambiguous, cells were discarded or scored visually using a cropped time-lapse. While NEBD is the definitive start of mitosis, we use the first time point after NEBD at which spindle length reaches a stable minimum as the "start" of mitosis for 2 reasons: (1) we find that NEBD is more difficult to unambiguously score from spindle length plots; and (2) the time between NEBD and what we define as the "start" of mitosis is relatively constant across strains and conditions and so not a major contributor to variation in mitotic timing. These mitotic events were used as landmarks to calculate the duration of mitosis, as described previously [23,24], and spindle features related to spindle size and dynamics.

## Analysis of GSPC spindle features

Spindle features were calculated in MATLAB. The following features were extracted:

A. Pole separation prior to NEBD: the mean of the top 3 spindle length measurements within 3 min prior to NEBD.

B. Duration of mitosis: as described previously.

C. Spindle length: mean spindle length between the "start" of mitosis ($t_s$) and anaphase onset ($t_{AO}$)

D. Spindle length variance: the standard deviation for a random sampling of 7 spindle length measurements between $t_s$ and $t_{AO}$, after spindle length was normalized to mean spindle length (as defined in C).

E. Spindle length fluctuation: the mean rate of change in spindle length, frame-to-frame, throughout mitosis, calculated as the change in spindle length between frames, as a percent of mean spindle length (as defined in C), divided by the frame rate.

F. Spindle rotation: the mean rate of spindle rotation, frame-to-frame, throughout mitosis, calculated as the angle between spindle vectors in adjacent frames divided by the frame rate (i.e. angular velocity).

G. Spindle elongation rate: the mean rate of change in spindle length between $t_{AO}$ and 2.5 min after $t_{AO}$, calculated as the slope of the line fit to spindle length versus time within this time window.

H. Spindle rotation anaphase: as in F, but measured between $t_{AO}$ and 2.5 min after $t_{AO}$.

I. Pole separation telophase: the second largest spindle length measured at least 7 min after $t_{AO}$ and expressed as a percent of mean spindle length (as defined in C).

Raw data are in S1 Data. Feature name abbreviations are preserved for compatibility with MATLAB scripts. Letters correspond to descriptions given above.

To generate the heatmap in S2B Fig, outliers were removed for all features except duration of mitosis. Outliers were defined as values that were more than 1.5x the interquartile range above the 75th quartile or below the 25th quartile. Outliers for each feature were assessed on a per genotype basis (e.g. outliers within the *daf-2(e1370)* data were defined based on the *daf-2 (e1370)* values). A Kruskal-Wallis test with a Tukey-Kramer post hoc test (MATLAB *kruskal-wallis* and *multcompare*) was performed, with a threshold for significance of $p < 0.05$, to determine whether any genotype was statistically different from control (shown as asterisks in the heatmap). The mean value for each feature for each genotype was normalized to control, and these normalized values (# standard deviations above/below control mean) were plotted as a heatmap (MATLAB *heatmap*). To determine whether any mitotic feature correlated with mitotic duration across all genotypes (S2C Fig), all measurements for a given feature were combined, standardized ($x-\mu/\sigma$), and tested for a correlative relationship with the duration of mitosis by assessing the Pearson's linear correlation coefficient (r) and $p$ value, after Bonferroni correction for multiple testing. Features showing a correlation with $p < 0.05$ were plotted as a heatmap as above. Non-significant correlations are shown as 0. Principal component analysis (S2D Fig) was performed after data standardization using the *pca* function in MATLAB. Data were plotted along the first two principal components, which account for ~45% of the total variance, and features were overlaid using the *biplot* function. Cells were classified as "delayed" if their duration of mitosis was greater than the 90th percentile of the control dataset published in [24]. Delayed cells from *daf-2(e1370)*, *daf-7(e1372)* and *glp-1(e2141)* are colored-coded accordingly.

## Measuring the number of mitotic GSPCs, the number of PZ nuclei and the PZ mitotic index

The number of mitotic GSPCs, number of PZ nuclei and the PZ mitotic index were measured in the same germ lines as were used for live-cell imaging. After XY registration, any germ lines in which the distal germ line was cut off in XY or Z were excluded. Mitotic cells were counted at the first frame, using the GFP::TBB-2 signal to identify all cells in which 2 spindle poles could be easily discerned (i.e. all cells in early prophase through early telophase). Mitotic cells were scored manually in Fiji using the point selection tool and saving the location of all scored cells as an ROI list. The total number of cells in the PZ was counted as the number of nuclei between the distal tip of the gonad and the first row of crescent shaped meiotic nuclei using the HIS-11::mCH signal [114,115]. Nuclei were counted from the first frame without appreciable animal movement. PZ nuclei counts were obtained by re-purposing the spot detection tools available in TrackMate [112,113]. After pre-processing the HIS-11::mCH channel (background subtraction, median 3D filtering with X and Y radii = 2 μm and Z radius = 1 μm, and gaussian blurring with sigma = 2) and manually defining a rectangular ROI encompassing the PZ, the Differences of the Gaussian (DoG) blob detector (estimated blob diameter = 4.25 μm) and the linear assignment problem (LAP) spot linker (maximum link and gap-closing distances = 3.0 μm, maximum gap allowance = 2 frames), were used to identify and track nuclei over at least 5 frames. Spots were manually curated to exclude non-nuclei spots (e.g. auto-fluorescent gut granules) or non-PZ nuclei (e.g. nuclei in the proximal gonad arm). The TrackMate 'Spots' information for each PZ was exported as a CSV file and the XYZ

coordinates for all nuclei were imported into MATLAB to extract nuclei counts, align nuclei counts with the respective number of mitotic cells and to calculate the mitotic index (number of mitotic cells/number of PZ nuclei). We note that our PZ nuclei counts for control strains are similar to reported values [30], while our mitotic indices are higher [29], likely owing to the fact that we can detect cells across more mitotic stages (prophase to telophase) than methods that count mitotic figures or phospho-Histone H3 positive cells.

## Measuring DAF-16::GFP nuclear enrichment

For single time point images, 9 interphase nuclei per germ line were selected manually using the HIS-11::mCH channel. A circular ROI, 4.5 μm in diameter, centered in XY and Z on each nucleus was drawn manually using the oval area selection tool. To generate cytoplasmic ROIs, the radius of each nuclear ROI was enlarged by 1 μm, and a donut-shaped composite ROI was generated using the XOR function in the ROI manager. Neighboring nuclei were outlined manually using the oval area selection tool and excluded from the cytoplasmic ROI using the XOR function (see Fig 2D). DAF-16::GFP fluorescence intensity (FI) was measured by applying these ROIs to the GFP channel and measuring the mean FI per pixel within each. After background subtraction (the mean FI per pixel across 3 rectangular ROIs positioned outside of the animal), the nuclear-to-cytoplasmic ratio for each cell was calculated as the mean nuclear FI divided by the mean cytoplasmic FI. For data shown in S5A Fig, where DAF-16::GFP was measured in animals carrying DAF-2::mNG::AID, an additional background correction was applied by measuring, in parallel, the germline FI of DAF-2::mNG::AID in animals carrying the same TIR1 drivers, but without DAF-16::GFP, as described below. These values were subtracted from the relevant DAF-16::GFP measurements (e.g. *sun1p*::*TIR1* DAF-2::mNG::AID 1mM Aux from *sun1p*::*TIR1* DAF-2::mNG::AID DAF-16::GFP 1mM Aux) prior to calculating the nuclear-to-cytoplasmic ratio. To measure DAF-16::GFP nuclear enrichment in timelapse images, animal movement was corrected by XY registration, as for spindle pole tracking. Mitotic nuclei were tracked manually by drawing a 4.5 μm diameter circular ROI, centered in XY and Z on each nucleus, over time. Cytoplasmic ROIs were generated and the DAF-16::GFP nuclear-to-cytoplasmic ratio was calculated for each time point as described above. The coordinates of the nuclear ROI were also used to generate an average intensity projection of 5 z-slices (0.75 μm step size), centered in Z on the tracked nucleus over time. Changes in the variance of the HIS-11::mCH FI within the nuclear ROI in these projections was used to estimate the duration of mitosis, as described previously [22]. Briefly, the HIS-11::mCH variance for each nucleus was min-max normalized, and plotted relative to time in MATLAB. NEBD was scored as the last time point when normalized variance was less than 50% of max and/or just prior to the sharp increase that characterizes the start of chromosome congression. Anaphase onset was scored as the last time point when normalized variance was greater than 70% of max and/or just prior to the sharp decrease that characterizes the start of chromosome segregation. Plots were scored using *ginput* in MATLAB. The time of NEBD was used to extract the DAF-16::GFP nuclear-to-cytoplasmic ratio prior to mitosis, by taking the mean of the highest 3 values measured at least 10 min prior to NEBD. 10 min was selected as the nuclear-to-cytoplasmic ratios measured at this time were similar to interphase nuclei. After 10 min, nuclear DAF-16::GFP decreases (See Fig 2E).

## Measuring DAF-2::AID::mNG and DAF-16::GFP::AID germ line depletion

2-channel, single time point Z-stacks were taken of the germ line and the middle z-slice for each germ line was selected using TBB-2::mCH signal. For DAF-2::AID::mNG measurements, a maximum intensity projection of 7 z-slices (0.75 μm step size), centered at this mid-plane,

was generated. For DAF-16::GFP::AID measurements, the mid-plane z-slice was used. Three rectangular ROIs (~9.7 x 12.9 μm) were drawn within the distal germ line region using the TBB-2::mCH signal and used to measure the mean FI. Background FI was measured by performing the same procedure for strains carrying the different TIR1 drivers and TBB-2::mCH alone. Germ lines with FI less than or equal to background were set to 0 and data were normalized as indicated in the figure legends.

## Scoring mitotic errors in GSPCs

Mitotic cells were tracked using either the TBB-2::GFP signal and spindle pole tracking, as described above, or the HIS-11::mCH signal, as described for counting PZ nuclei, followed by manual curation in TrackMate [112,113] to identify mitotic nuclei. Tracking coordinates were used to generate a cropped maximum intensity projection (11 z-slices, 0.5 μm step size) of each cell. Cropped cells were collected in a single folder and file names were replaced with an arbitrary number such that users were blind to the genotype of each cell during scoring. Cells were scored manually using a semi-automated script that opened each file and allowed users to score cells according to the following criteria: (1) no obvious defects; (2) misaligned chromosomes during prometaphase/metaphase; (3) lagging, bridging and/or misaligned chromosomes during anaphase.

## Measuring progeny production

L4 stage larvae from well fed NGM/OP50 plates were singled onto new NGM/OP50 plates and incubated at 20°C. Animals were transferred to new plates every 24 hrs for a total of 72 hrs. The total number of mature adults on each plate 72 hrs later was counted manually using a stereomicroscope. Animals that were sterile, dead, or bagged, were excluded from the analysis. Data were compiled in Excel.

## Measuring embryonic lethality

L4 stage larvae from well fed NGM/OP50 plates were transferred to a new NGM/OP50 plate and left to develop to adulthood for 24 hrs at 20°C. Synchronous 1-day adults were then singled onto new NGM/OP50 plates and allowed to lay eggs for 8hr at 20°C. Using a stereomicroscope, the total number of eggs laid were counted at the end of the 8hr collection. 24 hrs later the number of unhatched eggs were counted. Percent embryonic lethality was calculated as the number of unhatched eggs divided by the total number of eggs laid. Data were compiled in Excel.

## Counting germ line apoptotic bodies

L4 stage larvae expressing a CED-1::GFP fusion protein in the somatic sheath cells were transferred from well fed NGM/OP50 plates to fresh plates and allowed to develop for 24 hr at 20°C to reach the adult stage. 1-day adults were mounted as for live-cell imaging and single time-point z-stack images of the gonadal loop region were captured. Cell corpses were identified using the CED-1::GFP signal, which accumulates between dying cells and those that are engulfing them [116]. Gonads were scored manually, with genotype information removed (as for mitotic error scoring) to minimize user bias.

## Data availability, statistics and figure assembly

Statistical analyses and data plotting were performed in MATLAB. Summary statistics for all experiments, including statistical tests used, are provided in S2 Data. Numeric data for S2 Fig

are provided in S1 Data. Imaging data for the candidate screen described in Figs 1, S1 and S2 have been deposited at the Federated Research Data Repository (FRDR) [117]. Numeric data for all figure panels have been deposited at Dryad Digital Repository [118]. Figures were assembled using Adobe Illustrator. For all plots showing GSPC mitotic duration, small dots represent individual GSPCs, larger dots represent the mean duration of mitosis per gonad arm/PZ. For all other plots, dots represent one gonad arm/PZ and/or one animal. For all experiments, one gonad arm was assessed per animal. Boxplots show the median, interquartile range and most extreme values not considered statistical outliers. Bar plots show the mean with error bars showing the standard deviation. n.s. = $p > 0.05$, * = $p < 0.05$, ** = $p < 0.01$, *** = $p < 0.001$. All representative images were scaled to the same brightness and contrast settings, pseudo colored and cropped in Fiji, before exporting as RGB TIFs. Rotated images are indicated by the inclusion of background-colored pixels. Images were re-sized and/or masked in Adobe Illustrator to fit the final figure panel.

## Supporting information

**S1 Fig. Related to Fig 1.** (A) A maximum intensity projection through the top portion of the distal region of one gonad arm from an L4 larva showing germ cell nuclei marked by HIS-11::mCH. The PZ is outlined, with an arrow indicating a crescent-shaped meiotic nucleus marking the proximal end of the PZ. Scale bar = 10 μm. The number of nuclei per PZ used to calculate the mitotic index (Fig 1D) is shown below. (B) A maximum intensity projection through the top portion of the distal region of the same gonad arm shown in (A) showing the GFP::TBB-2 signal. Arrow heads show examples of mitotic spindles in different stages of mitosis (prophase through telophase). The number of mitotic cells per PZ used to calculate the mitotic index (Fig 1D) is shown below. The dashed grey line indicates the median value for control. For all plots, dots represent one PZ and one PZ was assessed per animal. Boxplots show the median, interquartile range and most extreme values not considered statistical outliers. n.s. = $p > 0.05$, * = $p < 0.05$, ** = $p < 0.01$, *** = $p < 0.001$. Summary statistics and statistical tests used for all figure panels are given in S2 Data.
(PDF)

**S2 Fig. Related to Fig 1.** (A) Schematic showing the spindle features extracted for GSPCs from animals bearing mutant alleles in the signaling pathways shown in Fig 1C. Features are shown relative to the spindle length versus time plot from Fig 1B, indicating which stage of mitosis they relate to. Features are listed on the right with letter designations used in (B-D). (B) Heatmap showing the mitotic features that differ significantly from control in each mutant background. Data were compared using a Kruskal-Wallis with Tukey-Kramer post hoc test with the null hypothesis that all samples come from the same distribution. Red indicates values above the control mean and blue indicates values below the control mean. Color saturation indicates the distance above/below the control mean after data standardization. (C) Heatmap showing the strength of the linear relationship between each spindle feature and the duration of mitosis across all cells of all genotypes. Pearson's coefficient (r) is shown for significant ($p < 0.05$ after Bonferroni correction) correlations. (D) Principal component analysis of spindle features showing all cells plotted along the first two principal components, with delayed cells (duration of mitosis > 90th percentile of control) shown for *daf-2(e1370)*, *daf-7(e1372)* and *glp-1(e2141)*. Vectors represent the contribution of each spindle feature to the principal components shown. All data used in this analysis can be found in S1 Data.
(PDF)

**S3 Fig. Related to Fig 2.** (A) The PZ mitotic index in animals bearing the mutant alleles or allele combinations indicated. *daf-18(nr2037)*, *akt-1(mg144)* and *daf-16(mu86)* rescue the lower mitotic index seen in *daf-2(e1370)* animals. Dots represent one PZ and one PZ was assessed per animal. (B) The DAF-16::GFP nuclear-to-cytoplasmic ratio (N:C) is elevated in GSPCs from animals in which *daf-2* was knocked down by RNAi. Small dots represent individual GSPCs, larger dots represent the mean value per gonad arm/PZ. (C) Single time point maximum intensity projections showing DAF-16::GFP nuclear localization over the course of a GSPC mitosis. Numbers indicate time in minutes relative to mitosis start, as inferred from HIS-11::mCH variance. Scale bar = 5 µm. For (A) and (B), boxplots show the median, interquartile range and most extreme values not considered statistical outliers. n.s. = $p > 0.05$, * = $p < 0.05$, *** = $p < 0.001$. Summary statistics and statistical tests used for all figure panels are given in S2 Data.
(PDF)

**S4 Fig. Related to Fig 3.** (A) Maximum intensity projection of the distal germ line of a late L4 larvae showing DAF-2::AID::mNG (green) and germ cell membranes (magenta; tagRFP::PH). DAF-2::AID::mNG is shown in inverted grey scale below. DAF-2::AID::mNG in the distal germ line is predominantly found in cytoplasmic puncta (arrow heads) in GSPCs and in the rachis (the shared inner core of cytoplasm to which all germ cells are connected via cytoplasmic bridges). Scale bar = 10 µm. (B) The mean DAF-2::AID::mNG fluorescence intensity per distal germ line, normalized to control animals (no TIR1). Dots represent the mean value per each gonad arm and one gonad arm was assessed per animal. Bar plots show the mean with error bars showing the standard deviation. DAF-2::AID::mNG levels are lower in *sun-1p*::TIR1 animals without auxin treatment. DAF-2::AID::mNG levels in *eft-3p*::TIR1 germ lines without auxin treatment are not different from control, but are higher compared to *sun-1p*::TIR1 germ lines without auxin. Auxin treatment does not deplete DAF-2::AID::mNG in *lim-7p*::TIR1 or *ges-1p*::TIR1 germ lines, with slightly elevated levels in *ges-1p*::TIR1 germ lines following auxin treatment. (C) Maximum intensity projections showing DAF-2::AID::mNG in the germ line (outlined in orange) and developing vulva (blue arrow head) in control animals (no TIR1) and in animals carrying *sun-1p*::TIR1 and *eft-3p*::TIR1 without auxin treatment, and in *lim-7p*::TIR1 and *ges-1p*::TIR1 with auxin treatment. The vulva is shown as an example of somatic depletion in *eft-3p*::TIR1. Scale bar = 10 µm. (D) GSPC mitosis is delayed in animals after *daf-2* was depleted by RNAi throughout the whole worm but not when *daf-2* was depleted in the germ line alone. Small dots represent individual GSPCs, larger dots represent the mean value per gonad arm/PZ. Boxplots show the median, interquartile range and most extreme values not considered statistical outliers. (E) Lower magnification, whole worm images showing partial DAF-2::AID::mNG depletion in *eft-3p*::TIR1 animals without auxin treatment. Orange arrow heads indicate prominent sites of somatic DAF-2::AID::mNG expression (head neurons, vulva, intestine) in control animals (no TIR1; left) and the same anatomical locations in animals bearing *eft-3p*::TIR1 showing reduced expression. Scale bar = 50 µm. For all plots, n.s. = $p > 0.05$, * = $p < 0.05$, *** = $p < 0.001$. Summary statistics and statistical tests used for all figure panels are given in S2 Data.
(PDF)

**S5 Fig. Related to Fig 3.** (A) The GSPC DAF-16::GFP nuclear-to-cytoplasmic ratio (N:C) in control animals (no TIR1), *sun-1p*::TIR1 animals with and without auxin, and *eft-3p*::TIR1 animals without auxin. The DAF-16::GFP N:C is elevated in *sun-1p*::TIR1 and *eft-3p*::TIR1 GSPCs without auxin treatment, compared to control, but increases further in *sun-1p*::TIR1 animals after auxin treatment. Small dots represent individual GSPCs, larger dots represent the mean value per gonad arm/PZ. Boxplots show the median, interquartile range and most extreme

values not considered statistical outliers. (B) Single z-slice sections through the middle of the germ line (orange outline; left), or a basal region through intestinal nuclei (blue arrow heads; right) in the same animals, showing DAF-16::GFP::AID expression in *daf-2*(RNAi) treated animals. As with DAF-2::AID::mNG, *eft-3p*::*TIR1* leads to partial somatic depletion of DAF-16::GFP::AID without the addition of auxin (compare top intestinal nuclei to those in either of the bottom 2 images). Auxin treatment leads to nearly complete DAF-16::GFP::AID depletion, in both germ line and soma, in *eft-3p*::*TIR1* animals, and robust germ line depletion in *sun-1p*::*TIR1* animals. Scale bar = 20 μm. Quantification of germ line DAF-16::GFP::AID fluorescence for a subset of the animals assayed in Fig 3E is shown on the right. Dots represent the mean value per each gonad arm and one gonad arm was assessed per animal. Bar plots show the mean with error bars showing the standard deviation. For all plots, n.s. = $p > 0.05$, * = $p < 0.05$, ** = $p < 0.01$, *** = $p < 0.001$. Summary statistics and statistical tests used for all figure panels are given in S2 Data.
(PDF)

**S6 Fig. *daf-2(e1370)* delays GSPC mitosis but does not impact the PZ mitotic index in 1-Day adults.** (A-B) The duration of mitosis (A), and the PZ mitotic index (B) of 1-Day adult control and *daf-2(e1370)* mutant animals. In *daf-2(e1370)* 1-Day adults, mitosis is delayed but there is no difference in the mitotic index. In (A), small dots represent individual GSPCs, larger dots represent the mean value per gonad arm/PZ. In (B), dots represent one PZ and one PZ was assessed per animal. In both (A) and (B), boxplots show the median, interquartile range and most extreme values not considered statistical outliers. n.s. = $p > 0.05$, *** = $p < 0.001$. Summary statistics and statistical tests used for all figure panels are given in S2 Data.
(PDF)

**S1 Table. Strain list.** Unless otherwise indicated, all strains were generated as part of this study, by standard genetic crosses. The source strains for alleles and transgenes are indicated the first time a strain carrying them appears in the table.
(DOCX)

**S2 Table. Imaging parameters.** Imaging parameters are listed by experiment type with relevant figures indicated. Imaging parameters were held constant across experiments with occasional exceptions. Where multiple settings were used, the most common are in bold. For spindle pole and mitotic error tracking, variation in imaging parameters was within the range commonly used by our lab and below the threshold for phototoxicity as determined in [24]. For experiments quantifying fluorescence intensity, data were normalized to the same point of reference to permit comparison across experiments.
(DOCX)

**S1 Data. Candidate screen spindle features.**
(XLSX)

**S2 Data. Summary statistics for all figures.**
(XLSX)

## Acknowledgments

We thank Drs. R. Roy (McGill), C. Rocheleau (McGill), S. Weber (McGill) and J.-C. Labbé (IRIC, UdeM), members of the Gerhold and Weber labs, and members of the Montreal Area Worm Meeting group for helpful feedback and sharing protocols, reagents and strains. We are

also grateful to Drs. P. Lara-Gonzalez (UC Irvine), A. Desai (UCSD) and B. Lacroix (CRBM) for sharing strains. Some strains were provided by the CGC, which is funded by NIH Office of Research Infrastructure Programs (P40 OD010440). We thank Wormbase [119], which was used regularly throughout this project, and our funding sources.

## Author Contributions

**Conceptualization:** Abigail R. Gerhold.

**Data curation:** Eric Cheng, Abigail R. Gerhold.

**Formal analysis:** Eric Cheng, Ran Lu, Abigail R. Gerhold.

**Funding acquisition:** Abigail R. Gerhold.

**Investigation:** Eric Cheng, Ran Lu.

**Methodology:** Eric Cheng, Ran Lu, Abigail R. Gerhold.

**Project administration:** Abigail R. Gerhold.

**Resources:** Abigail R. Gerhold.

**Software:** Abigail R. Gerhold.

**Supervision:** Abigail R. Gerhold.

**Validation:** Eric Cheng, Abigail R. Gerhold.

**Visualization:** Eric Cheng, Abigail R. Gerhold.

**Writing – original draft:** Eric Cheng, Abigail R. Gerhold.

**Writing – review & editing:** Eric Cheng, Ran Lu, Abigail R. Gerhold.

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
