## [Decision Letter · Decision Letter 0]

5 Aug 2024

Dear Dr Gerhold,

Thank you very much for submitting your Research Article entitled 'Non-autonomous insulin signaling regulates the duration of mitosis in *C. elegans* germline stem and progenitor cells' to PLOS Genetics.

The manuscript was fully evaluated at the editorial level and by independent peer reviewers. The reviewers appreciated the attention to an important problem, but raised some substantial concerns about the current manuscript. Based on the reviews, we will not be able to accept this version of the manuscript, but we would be willing to review a much-revised version. We cannot, of course, promise publication at that time.

If you decide to revise the manuscript for further consideration at PLOS Genetics, please aim to resubmit within the next 60 days, unless it will take extra time to address the concerns of the reviewers, in which case we would appreciate an expected resubmission date by email to plosgenetics@plos.org.

To resubmit, log into your Editorial Manager account and select the option 'Revise Submission' in the 'Submissions Needing Revision' folder.

We are sorry that we cannot be more positive about your manuscript at this stage. Please do not hesitate to contact us if you have any concerns or questions.

Yours sincerely,

Sean P. Curran

Academic Editor

PLOS Genetics

Pablo Wappner

Section Editor

PLOS Genetics

Reviewer's Responses to Questions

**Comments to the Authors:**

Reviewer #1: Summary/Critique

In this manuscript, Cheng and colleagues investigate the role of the DAF-2 insulin/IGF receptor and key downstream signaling pathway components (e.g, DAF-16/FOXO, DAF-18/PTEN, and AKT-1) in influencing the behavior of germline stem/progenitor cells during the L4 larval stage. It was know that daf-2 mutants exhibit a lower mitotic index during larval (but not adult) stages (Michaelson et al., 2010). It was found that the response to insulin signaling occurs cell autonomously in the germline to promote the proliferation of germline stem/progenitor cells (Michaelson et al., 2010). The slower rate of germline proliferation in larval stages in daf-2 mutants was attributed to delays in progression through the G2 phase of the cell cycle (Michaelson et al., 2010). In this new work, Cheng and colleagues use live cell recording during the the L4 stage and show that the duration of M phase is extended by about 3 minutes on average (with a range of up to about 20 minutes) in daf-2 mutants. In contrast to the extensive delay in larval germline proliferation, which requires DAF-2 function in the germline (and ins-3 and ins-33 in the soma), the observed increase in the duration of mitosis involves the somatic action of DAF-2. I found the experiments to be carefully done and well described. The auxin-mediated depletion studies are nicely done. The manuscript is well written and the many statistical tests employed appear to be appropriate. The results support the authors's conclusions. The authors might consider these specific points.

1. What is the biological significance of the observations here? Germline proliferation during larval stages is already reduced in daf-2 mutants owing to elongation of the cell cycle, which appears largely to be due to an extension of the G2 phase of the cell cycle (Michaelson et al., 2010). Given that the mitotic cell cycle in the germline has a median length of approximately 2.5 hours in the wild type (Fox et al., 2011), what would be the impact of the extensions of M phase observed here? Addressing this point would help general readers better appreciate the biological relevance of this study.

2. A concern is that that physiology of daf-2 mutants makes them more sensitive to the conditions used for the time-lapse recordings (e.g., absence of food, compression, phototoxicity). Given that the mitotic index of daf-2 mutants was not reported to be affected in the adult stage (Michaelson et al., 2010), I wonder whether the authors would observe similar extensions of M phase in the adult stage. If not, this would a good control and might provide an additional contrast to the prior work. Further, in their Discussion section, the authors speculate on potential mechanisms. On the face of it, the mechanisms they envision and discuss would not seem to be limited exclusively to larval stages.

3. I'm not sure it is fair to say here that DAF-2 insulin signaling regulates germline stem/progenitor cells (e.g., in the Title and Abstract). Rather, the results simply indicate that germline stem/progenitor cells are affected in the mutant backgrounds in that the duration of M phase is extended.

4. Table S1. Strains carrying bcIs39 were generated with the lin-15(+) marker (not lim-15).

David Greenstein

Reviewer #2: Regulation of the stem and progenitor cell cell cycle in vivo is challenging to study. In principle stem and progenitor cell division should respond to the physiological needs of the organism but this hypothesis has only recently been investigated. Here Cheng and colleagues describe signaling pathways which regulate the duration of mitosis in the C. elegans germline stem and progenitor cell system (the progenitor zone, PZ), in late L4, near the end of the PZ expansion period to form the adult germline. Through a candidate screen of known mutations that affect stem cells and/or the PZ, the authors find that reduction of insulin/IGF, Notch, and DAF-7/TGF-beta pathways delay the duration of mitosis by live imaging experiments. The authors find that the insulin/IGF receptor, DAF-2, regulates mitosis through the canonical DAF-16/FoxO transcription factor and that the pathway works cell non-autonomously with at least the gonadal sheath and intestine being major sites of action. Mitotic delays with daf-2 loss-of-function are dependent on functioning spindle assembly checkpoint machinery but daf-2 reduction does not affect the prevalence of mitotic errors found in spindle assembly checkpoint mutants. Caloric restriction enhances the rate of mitotic errors in spindle assembly checkpoint mutants and delays the duration of mitosis. But unlike daf-2 mutants, caloric restriction appears to work through DAF-18/PTEN and AMPK but not the canonical insulin/IGF pathway. In general, the experiments are well executed and the data convincing. The finding that DAF-2 acts non-autonomously during larval expansion is exciting and consistent with it acting non-autonomously in adult aging (Qin and Hubbard, 2015).

Major points

1) The significant weakness of the manuscript is that the biological significance of the finding reported here for regulation of the mitotic cell cycle in the C. elegans stem-progenitor zone is unclear/not well explained. The current manuscript thus misleads the readership of PLoS Genetics into thinking that this work explains known germline development/physiological effects and signaling gene mutant phenotypes. The unaddressed question is – does the observed mitotic delays in the reported condition explain the germline biology that is observed in these conditions, as implied by Figure 1B and text. The authors must cover these issues in the Discussion, at least.

Published work of others showed that the cell cycle time (S phase, G2 and M) in wild type young adults is ~6.5 hr. The cell cycle time in late larval development is unknown and is thought to be faster than in adults, but is clearly multiple hours long (liberal estimate, 4 hrs). In this study, samples were examined in late L4 where the biology is that this time is nearing the end of the larval expansion of the PZ to yield the young adult PZ size. Changes in cell cycle time need to be considered and explained in the context of the germline biology and signaling that at least some of the readership of PLoS Genetics will know.

Cheng et al. rigorously show that in late L4, during larval expansion of the PZ, the duration of mitosis in daf-2(rf) is increased by a median of 2.2 min (Fig 1F) or 4.4 min per cell (Fig 2B) [average median increase of 3.3 min per cell].

(A) Published work of others showed that daf-2(rf) has fewer cells in the young adult PZ and has a reduced mitotic index. Time course studies show that there are fewer PZ cells in the adult (~130 cells in daf-2(rf)) compared to ~220 cells in wild type) is a result of reduce expansion of the PZ pool from early L3 to young adult (Michaelson et al., 2009).

The biological question of interest is - does the increase in duration of mitosis fully explain the defect in larval expansion in daf-2(rf)? The manuscript, as written, implies that it does, but actually does not address this central point, which was provided as a motivation for the study (Fig 1B, text, citations). If we assume an L4 cell cycle time of 4 hours, an average 3.3 min median mitotic phase delay represents only a 1.4% increase in cell cycle length. Will a 1.4% increase in cell cycle length result in the production of 125 PZ cells in daf-2(rf) versus 220 PZ cells in wt during expansion from ~40 cells in early L3 to young adult (~20hrs)? This reviewer don’t know, but it opens the strong possibility that the increase in duration of mitosis does not explain or explains only part of the effect on daf-2 signaling caused by the e1370 mutation.

B) Cheng et al. rigorously show that in late L4 the duration of mitosis increased in time by a median of 1.7 min per cell in glp-1(rf) (Fig 1F). The audience of this manuscript may go away with the idea, even though not directly discussed, that the increase in duration of mitosis during larval development (asses in late L4) leads to the smaller PZ in young adult glp-1(rf). It is reasonable to assume that the cell cycle time of glp-1(rf) at 20oC is 6.5 hrs, given that wt and glp-1(rf) have the same length at 15oC (~9 hrs, Fox and Schedl, 2015). A 1.7 min delay in mitosis corresponds to 0.4% of the cell cycle. It seems unlikely that a 0.4% increase in cell cycle length explains the decreased number of PZ cells in young adult glp-1(rf). Instead, the more likely explanation is that it is the decreased number of stem cells in glp-1(rf), based on published work using staining for stem cell markers, and experiments showing that increasing the number of stem cells correspondingly increases the size of the PZ.

C) What aspect of germline biology during caloric restriction does the mitotic delay explain?

D) This raises the possibility that the observed effect on mitotic phase duration, in at least some of the genetic backgrounds/diet tested, is to counteract conditions that increase the frequency of mitotic chromosome segregation defects (possible biology explained), rather than to change cell cycle length and thus PZ dynamics as part of a process to modify timing and number of self/mated progeny produced.

E) Published work has shown that signaling and regulation of stem and progenitor cells during larval development, in the adult, and in the distinct diapauses can show significant differences. Therefore, the authors need to make it clear in the Abstract, Introduction, Results and the Discussion that this working is examining signaling during larval expansion of the PZ, at late L4. Furthermore, when describing different genes and signaling pathways, the authors should indicate in what stage(s) these studies were conducted.

Minor points:

2) The abbreviation NEBD is not defined on line 147.

3) It would be useful to discuss the specific mitotic features that seem to be important in the different genotypes in the paragraph starting on line 156 as the current description is vague.

4) In Figure 3 the authors use strains containing DAF-2::AID::mNG, TIR1::Ruby, mCH::TBB-2, and DAF-16::GFP. How are the two red and two green fluorescent proteins separated in these experiments?

5) Line 203. Having 25 min is gratuitous, given that it is clearly an outlier in the data. It is well known the live imaging of the distal germline without inducing arrest or other aberrant behaviors is challenging; significant time delayed outliers may arise from such live distal germline imaging issues.

6) In Figure 3D the authors indicate that germline degradation of DAF-2 has no effect on mitosis but there appears to be a statistically significant reduction in the duration from the figure. The authors should state this result in the text and at least speculate why they would observe this effect, or is it that while the result is statistically significant it is so small as not to be biologically significant?

7) In line 251 the authors state they cannot fully exclude the possibility the germline autonomous DAF-16 regulates mitotic timing, but they have provided no information in support of the idea. The authors should remove the statement or explain why this cannot be excluded.

8) In line 261 the authors state that the intestine and gonadal sheath are the primary sites of DAF-2 activity, but could other cell types also be at play that were not tested? The authors should edit this statement to not exclude the possibility that DAF-2 may also function elsewhere.

9) In lines 280 and 282 the authors describe comparisons between genotypes as “the same” when formally they are failing to reject the null hypothesis of equivalent frequencies. These descriptions should be changed to describe the comparisons as not significantly different.

10) In the paragraph starting on line 292, the relationship between the amount of mitotic germ cells and sterility is unclear. Could these be two entirely independent phenotypes? In Figure S5B,C the number of mitotic cells is reduced in daf-2 mutants but sterility is unchanged, suggesting they are unrelated phenotypes. Can the authors provide direct evidence that the amount of mitotic germ cells is necessary for fertility?

11) It would be good to go into more detail about which spindle features relate to which genotypes in the paragraph on line 352.

12) On line 408 the authors indicate that germline depletion of DAF-2 had “no effect” on the number of mitotic cells but it is more accurate to say there was no significant difference.

13) In line 666 GSPCs are described as GSCs.

14) All figure panel where quantitative data is shown need to have more information on the statistical tests, description of the graphical features, and n values for the groups in the legend.

15) In Figure 1B the worm gene name for TOR is missing.

16) Figure 2D needs a genotype label.

17) Plots showing DAF-16::GFP nuclear:cytoplasm should start at 0 on the y axis.

18) Figure 4C,D should show p-values between all the groups.

19) Figure 4F may be better represented as a stacked bar graph showing the proportion of gonads binned into the number of apoptotic bodies.

20) In Figure S2D, the points should be labeled as delayed mitotic cells.

21) An arrow should be added to Figure S3C to point out which nucleus to focus on.

22) The arrows in Figure S4D are not pointing at the intestinal nuclei.

Reviewer #3: The manuscript titled: “Non-autonomous insulin signaling regulates the duration of mitosis in C. elegans germline stem and progenitor cells,” by Cheng et al. investigates the role of signaling pathways in the progression through the cell cycle and proper chromosome segregation during mitosis. The authors carry out a candidate genetic screen for signaling pathway mutants that affect mitosis and describe a role for the insulin/IGF receptor DAF-2 in promoting germline stem progenitor cells mitotic timing. DAF-2 acts through the canonical pathway by inhibiting the DAF-16 transcription factor. The experiments are rigorous and convincing, although the major conclusions are seemingly in contradiction to observations previously reported by Michaelson et al. 2012. Michaelson et al. from the Hubbard lab concluded that DAF-2 signaling through the canonical pathway acted to inhibit DAF-16 and that the germline responded to insulin signaling to ensure robust germline progenitor stem cell proliferation. The focus of action in the previous paper by Michaelson et al were mainly obtained by rescue experiments, i.e. experiments that assessed sufficiency of daf-2. However in this manuscript, Cheng et al carried out necessity experiments using degron tagged loci of daf-16 (DAF-16::AID) to determine in which tissues these two genes function to regulate mitosis. Cheng et al determined that the predominant site of activity for DAF-16 was in somatic tissues, particularly in the intestine and somatic gonad. This result is conflicting with the reported findings in Michaelson et al., and thus would make this work important and of significant interest. Additionally, the report also extends the findings by the Hubbard lab showing that the nuclear localization of DAF-16 in the germ line, in daf-2 mutants, does not correlate with an effect on mitotic timing. Cheng et al. further showed that the reduced daf-2 mitotic delays are entirely dependent on an intact spindle assembly checkpoint, and that while reduced daf-2 results in checkpoint dependent mitotic delays, mitotic fidelity was not affected; no defects in spindle assembly were noted in animals with reduced function of daf-2. This result stands in contrast to the delay in mitosis after caloric restriction, where Cheng et al. find mitotic fidelity to be compromised. Thus, this work provides new information, and these results indicate that the canonical DAF-2 signaling does not mediate the effect of caloric restriction on germline stem cell mitosis. Cheng et al., also reports that the long lifespan of eat-2 loss of function mutants or in animals under caloric restriction also requires daf-18/PTEN but not daf-16/PTEN. Another conflicting finding in the manuscript is that they do not see a phenotype in rsks-1 mutants, in conflict with the report from Korta et al. 2012. Overall, this is a well written article that reports potentially interesting if provocative findings. The reported findings should be of great interest for the community, but the manuscript in its current form fails to appropriately address the contradictions, either textually in an extended discussion or through additional experiments to explain the discrepancies.

Major points

1. The authors employ AID with eft-3 and lim-7::TIR1 strains. How was the eft-3 strain validated to ensure knock down only in somatic tissues with no effects on the germ line? Along the same lines, to my knowledge the lim-7::TIR1 has not been validated to make sure that only sheath cells show AID degradation. This has to be rigorously addressed to validate the provocative results.

2. The authors state in the Methods section that for live-cell imaging, larvae were raised on standard NGM plates seeded with E. coli strain HT115, which produces more uniform developmental timing and a more consistently high number of GSPC divisions. However, the Han lab (Chi et al., Genes and Development 2016) has shown that the commonly used HT115 E. coli strain carries a mutation in cytR and that this mutation makes the strain rich in nucleosides U/T. We do not know if this strain affects all mutants similarly. It strongly suppressed the phenotype of a double mutant of cdd-1 and cdd-2, two cytidine deaminases (Chi et al., Genes and Development 2016). It is possible that specific mutants are rescued by the feeding of this strain rather than others, or their phenotype enhanced. This could certainly explain the conflicting lack of phenotype of rsks-1 mutants also observed in this report. Thus, for live-cell imaging, the authors should still show the results of the analysis with OP50 E. coli.

3. A major finding of the report is that DAF-2 and DAF-16 are required in somatic tissues, such as intestine and sheath cells for the timing of the germline mitotic divisions. However, Michaelson et al. had provided rescue experiments and mosaic analysis to show that DAF-16 was required in the germline. Their conclusion was that daf-2, daf-16, daf-18, and hcf-1 were required in the germline primarily, “with a possible additional minor contribution from the soma.” In addition, Michaelson et al reported that the ablation of the SS cells, the precursor cells to the sheath and spermatheca, in daf-16(+) and daf-16(-), by RNAi, made no difference, suggesting that DAF-16 in the sheath cells was not a source that promoted germline proliferation. An interesting finding from this report by Cheng et al is that nuclear enrichment of DAF-16/FOXO in the germ line did not appear to affect mitotic timing. These results from Cheng et al. are provocative, however at the very least, they need to expand their discussion of the conflicting results and provide some possible explanations.

4. The authors describe that strains carrying TIR1 expressed from different promoters and the DAF-2::AID::mNG entered dauer when raised from L1 on 1mM K-NAA plates, but that animals raised in control (no K-NAA) showed a “weak daf-2 phenotype.” Have the authors tested other existing DAF-2::AID strains? There are at least two other strains available in the literature [Venz et al., Elife 2022; Roy et al., Aging Cell 2022]. A different AID-degron tagged strain would perhaps allow for the analysis of DAF-2.

5. In the methods, the authors describe that synchronized animals were obtained by hypochlorite treatment, however this may introduce added stress. Are the authors sure that hypochlorite treatment has no effect on the process they study? A better way to synchronize animals is to gather L1 as they hatch off, see McCarter et al., 1997.

Minor points:

On page 8, line 147, the authors should define NEBD. They do define it in the figure legends, but this is the first time NEBD is used in the body of the results section.

**Have all data underlying the figures and results presented in the manuscript been provided?**

Reviewer #1: Yes

Reviewer #2: Yes

Reviewer #3: None

PLOS authors have the option to publish the peer review history of their article (what does this mean?). If published, this will include your full peer review and any attached files.

Reviewer #1: **Yes: **David Greenstein

Reviewer #2: No

Reviewer #3: No

---

## [Decision Letter · Decision Letter 1]

25 Nov 2024

Dear Dr Gerhold,

We are pleased to inform you that your manuscript entitled "Non-autonomous insulin signaling delays mitotic progression in C. elegans germline stem and progenitor cells" has been editorially accepted for publication in PLOS Genetics. Congratulations!

Yours sincerely,

Sean P. Curran

Academic Editor

PLOS Genetics

Pablo Wappner

Section Editor

PLOS Genetics

Aimée Dudley

Editor-in-Chief

PLOS Genetics

Anne Goriely

Editor-in-Chief

PLOS Genetics

Comments from the reviewers (if applicable):

Reviewer's Responses to Questions

**Comments to the Authors:**

Reviewer #2: The authors have done an excellent job of addressing the concerns.

Reviewer #3: The manuscript titled now: “Stem cell mitosis in vivo and its regulation by signaling pathways,” by Chen et al. investigates the role of signaling pathways in the progression through the cell cycle and proper chromosome segregation in mitosis. This is a well written article where the authors carry out a candidate genetic screen for signaling pathway mutants that affect mitosis and describe a role for the insulin/IGF receptor DAF-2 in promoting germline stem progenitor cells mitotic timing. DAF-2 acts through the canonical pathway and the DAF-16 transcription factor. The authors have addressed all of my concerns. I find the manuscript much improved. The authors have clarified all of the comments I had on the previous manuscript and have added new important data. In summary, I think that the conclusion that the insulin pathway is acting cell non-autonomously to influence germline stem and progenitor cell mitosis is well supported.

**Have all data underlying the figures and results presented in the manuscript been provided?**

Reviewer #2: Yes

Reviewer #3: Yes

PLOS authors have the option to publish the peer review history of their article (what does this mean?). If published, this will include your full peer review and any attached files.

Reviewer #2: No

Reviewer #3: No

**Data Deposition**

http://datadryad.org/submit?journalID=pgenetics&manu=PGENETICS-D-24-00712R1

**Press Queries**

---

## [Editor Report · Acceptance letter]

12 Dec 2024

PGENETICS-D-24-00712R1 

Non-autonomous insulin signaling delays mitotic progression in C. elegans germline stem and progenitor cells 

Dear Dr Gerhold, 

We are pleased to inform you that your manuscript entitled "Non-autonomous insulin signaling delays mitotic progression in C. elegans germline stem and progenitor cells" has been formally accepted for publication in PLOS Genetics! Your manuscript is now with our production department and you will be notified of the publication date in due course.

With kind regards,

Anita Estes

PLOS Genetics

On behalf of:
